# OFFLINE EQUILIBRIUM FINDING

## ABSTRACT

Offline reinforcement learning (Offline RL) is an emerging field that has recently begun gaining attention across various application domains due to its ability to learn behavior from earlier collected datasets. Offline RL proved very successful, paving a path to solving previously intractable real-world problems, and we aim to generalize this paradigm to a multi-agent or multiplayer-game setting. To this end, we formally introduce a problem of *offline equilibrium finding* (OEF) and construct multiple datasets across a wide range of games using several established methods. To solve the OEF problem, we design a model-based method that can directly apply any online equilibrium finding algorithm to the OEF setting while making minimal changes. We focus on three most prominent contemporary online equilibrium finding algorithms and adapt them to the OEF setting, creating three model-based variants: OEF-PSRO and OEF-CFR, which generalize the widely-used algorithms PSRO and Deep CFR to compute Nash equilibria (NEs), and OEF-JPSRO, which generalizes the JPSRO to calculate (Coarse) Correlated equilibria ((C)CEs). We further improve their performance by combining the behavior cloning policy with the model-based policy. Extensive experimental results demonstrate the superiority of our approach over multiple model-based and model-free offline RL algorithms and the necessity of the model-based method for solving OEF problems. We hope that our efforts may help to accelerate research in large-scale equilibrium finding.

## 1 INTRODUCTION

Game theory provides a universal framework for modeling interactions among cooperative and competitive players (Shoham & Leyton-Brown, 2008). The canonical solution concept is Nash equilibrium (NE), describing a situation when no player increases their utility by unilaterally deviating. However, computing NE in two-player or multi-player general-sum games is PPAD-complete (Daskalakis et al., 2006; Chen & Deng, 2006), which makes solving games both exactly and approximately difficult. The situation remains non-trivial even in two-player zero-sum games, no matter whether the players may perceive the state of the game perfectly (e.g., in Go (Silver et al., 2016)) or imperfectly (e.g., in poker (Brown & Sandholm, 2018) or StarCraft II (Vinyals et al., 2019)). In recent years, learning algorithms have demonstrated their superiority in solving large-scale imperfect-information extensive-form games over traditional optimization methods, including linear or nonlinear programs. The most successful learning algorithms belong either to the line of research on counterfactual regret minimization (CFR) (Brown & Sandholm, 2018), or policy space response oracles (PSRO) (Lanctot et al., 2017). CFR is an iterative algorithm approximating NEs using repeated self-play. Several sampling-based CFR variants (Lanctot et al., 2009; Gibson et al., 2012) were proposed to solve large games efficiently. To scale up to even larger games, CFR could be embedded with neural network function approximation (Brown et al., 2019; Steinberger, 2019; Li et al., 2019; Agarwal et al., 2020). The other algorithm, PSRO, generalizes the double oracle method (McMahan et al., 2003; Bošanský et al., 2014) by incorporating (deep) reinforcement learning (RL) methods as a best-response oracle (Lanctot et al., 2017; Muller et al., 2019). The neural fictitious self-play (NFSP) can be seen as a special case of PSRO (Heinrich et al., 2015). Both CFR and PSRO achieved great performance in solving large-scale imperfect-information games, e.g., poker (Brown & Sandholm, 2018; McAleer et al., 2020). In this paper, we only focus on the imperfect-information extensive-form games.

One of the critical components of these successes is the existence of *efficient and accurate simulators*. A simulator serves as an environment that allows an agent to collect millions to billions of trajectories for the training process. The simulator may be encoded using rules as in different poker variants (Lanctot et al., 2019), or a video-game suite like StarCraft II (Vinyals et al., 2017). However, in many real-world games such as football (Kurach et al., 2020; Tuyls et al., 2021) or table tennis (Ji et al., 2021), constructing a sufficiently accurate simulator may not be feasible because of a plethora of complex factors affecting the game-play. These factors include the relevant laws of physics, environmental circumstances (e.g., wind speed), or physiological limits of (human) bodies rendering certain actions unattainable. Therefore, the football teams or the table tennis players may resort to watching previous matches to alter their strategies, which semantically corresponds to *offline equilibrium finding* (OEF). Recent years have witnessed several (often domain-specific) attempts to formalize offline learning in the context of games. For example, the StarCraft II Unplugged (Mathieu et al., 2021) offers a dataset of human game-plays in this two-player zero-sum symmetric game. A concurrent work (Cui & Du, 2022) investigates the necessary properties of offline datasets of two-player zero-sum games to successfully infer their NEs.

However, neither work considers the significantly more challenging field of multi-player games. To this end, we propose a general problem – *offline equilibrium finding* (OEF) which aims to find the equilibrium strategy of the underlying game, given a fixed dataset collected by an unknown behavior strategy. It is a big challenge since it needs to build the relationship between an equilibrium strategy and an offline dataset. To solve this problem, we introduce an environment model as the intermediary between them. More specifically, our main contributions include i) proposing a novel problem – OEF, and constructing OEF datasets from widely accepted game domains using different behavior strategies; ii) proposing a model-based method that can generalize any online equilibrium finding algorithm to the OEF setting by introducing an environment model; iii) adapting several existing online equilibrium finding algorithms to the OEF setting for computing different equilibrium solutions; iv) applying the behavior cloning technique to further improve the performance; v) conducting extensive experiments to verify the effectiveness of our algorithms. The experimental results substantiate the superiority of our method over model-based and model-free offline RL algorithms and the effectiveness and the necessity of the model-based method for solving the OEF problem.

## 2 RATIONALE BEHIND OFFLINE EQUILIBRIUM FINDING

We begin by providing a motivating scenario of OEF and comparing it to three related lines of research – opponent modeling, empirical game theoretic analysis, and offline RL – in order to highlight the rationale behind the introduction of OEF. We then justify the choice of model-based methods for OEF. A more detailed overview of related works is provided in Appendix A.

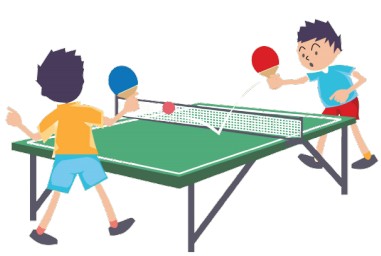

Figure 1: The game of table tennis.

**Motivating scenario.** Assume that a table tennis player $A$ will play against player $B$ whom they never faced before. What may $A$ do to prepare for the match? In this case, even though $A$ knows the rules of table tennis, they remain unaware of specific circumstances of playing against $B$, such as which moves or actions $B$ prefers or their subjective payoff function. Without this detailed game information, self-play or other online equilibrium finding algorithms cannot work. If player $A$ simply play the best response strategy against player $B$'s previous strategy, the best response strategy may be exploited if player $B$ change his strategy. Therefore, player $A$ has to *watch the matches that player $B$ played against other players* to learn their style and compute the equilibrium strategy, which minimizes exploitation. This process corresponds to the proposed OEF methodology.

**Why offline equilibrium finding?** In games with complex dynamics like table tennis games or football games (Kurach et al., 2020), it is difficult to build a realistic simulator or learn the policy during playing the game. A remedy is to learn the policy from the historical game data. Therefore, we propose the *offline equilibrium finding* (OEF) problem, which can be defined as

> *Given a fixed dataset $\mathcal{D}$ collected by an unknown behavior strategy $\sigma$, find an equilibrium strategy profile $\sigma^*$ of the underlying game.*

The OEF is similar to Offline RL but poses several unique challenges: i) the canonical solution concept is the game-theoretic mixed Nash equilibrium, which requires an iterative procedure of computing responses; ii) the games have at least two players who play against each other, which increases sensitivity to distribution shift and other uncertainties when compared with traditional Offline RL; and iii) the distribution shifts of the opponents' actions and the game are coupled, which brings difficulties to distinguish and address them.

**Why Opponent Modeling (OM) is not enough?** Opponent modeling is used to predict the opponents' behavior strategies, in both single and multi-agent reinforcement learning (He et al., 2016). However, predicting the opponents' behaviors only is not enough since the opponents in the OEF setting are not fixed but always best responding to the agent's strategy. In this work, we apply imitation learning, which is also used in opponent modeling, to learn the behavior cloning policy, and the experimental results show that simply applying the behavior cloning technique is not enough.

**Why does Empirical Game Theoretic Analysis (EGTA) cannot work?** The EGTA was proposed to reduce the complexity of large economic systems in electronic commerce (Wellman, 2006). It evolved into two main directions: strategic reasoning for simulation-based games (Wellman, 2006) and evolutionary dynamical analysis of agent behavior inspired by evolutionary game theory (Tuyls et al., 2018). The basic idea of EGTA is to use a sampled strategy to interact with a game simulator and estimate the empirical game from the simulation's results. It should consequently provide insights into the structure of the original game. PSRO (Lanctot et al., 2017) is a specific case of EGTA. In an offline equilibrium finding problem, only an offline dataset collected by an unknown strategy is available, while the game simulator and sampled strategy are not provided. Therefore, EGTA cannot be applied to find the equilibrium strategy only based on the offline dataset. Therefore, we propose a new model-based approach to find the equilibrium from an offline dataset. Our algorithm only relies on an offline dataset and does not require a game simulator or any known strategy.

**Why naive Offline RL is not enough?** The offline RL aims to learn a good behavior policy from previously collected datasets which can achieve the highest utility for the agent (Levine et al., 2020). However, maximizing the utility of each player in the game independently is not sufficient or practical, because the utility of one player depends not only on its action but also on the actions of other players. Therefore, if we use the offline RL algorithm to learn the best strategy for each player in the game, it may cause a huge loss since the computed best strategies for players may be exploitable. In other words, other players may play their best response strategies against the computed best strategy for the player instead of their behavior strategies in the offline dataset. Therefore, to compute the equilibrium strategy of the game, offline RL is not enough since it can only learn the best strategy for one agent independently. We also substantiate this claim empirically in the experiment section.

**Why model-based methods?** In traditional offline RL, the data of two actions may be used to determine which action is better (Levine et al., 2020). In contrast, the data of two action tuples in OEF are not enough to decide which tuple is closer to an equilibrium strategy as equilibrium identification requires other action tuples to serve as references (Cui & Du, 2022). For computing equilibrium strategies, we hence need to be able to evaluate any action tuple, and the data of some may be missing from the dataset. It prevents us from learning using model-free methods from the dataset directly. Therefore, we have to resort to simulating the game dynamics by a model to give the agents their utilities. In this context, *model-based methods, rather than other model-free methods*, are arguably the most appropriate choices for solving the OEF problem. Unfortunately, most known model-based methods for single-agent reinforcement learning cannot be directly applied to games for equilibrium finding. The main reason is their inherent reliance on the existence of no strategic opponents in the environment. Though MuZero (Schrittwieser et al., 2020) can be applied to symmetric two-player zero-sum games (e.g., Go) and solve the game with self-play, for heterogeneous players, it requires either a model for each agent separately or a model for all agents. Training these agents' models is challenging in OEF due to the limited game data.

## 3 PRELIMINARIES

### 3.1 IMPERFECT-INFORMATION GAMES

An imperfect-information game (IIG) (Shoham & Leyton-Brown, 2008) is represented as a tuple $(N, H, A, P, \mathcal{I}, u)$, where $N = \{1, ..., n\}$ is a set of players and $H$ is a set of histories (i.e., the

possible action sequences). The empty sequence $\emptyset$ corresponds to a unique root node of a game tree included in $H$, and every prefix of a sequence in $H$ is also in $H$. $Z \subset H$ is the set of the terminal histories. $A(h) = \{a : (h, a) \in H\}$ is the set of available actions at any non-terminal history $h \in H$. $P$ is the player function. $P(h)$ is the player who takes an action at the history $h$, i.e., $P(h) \mapsto N \cup \{c\}$. $c$ denotes the "chance player", which represents stochastic events outside of the players' controls. If $P(h) = c$ then chance determines the action taken at history $h$. Information sets $\mathcal{I}_i$ form a partition over histories $h$ where player $i \in N$ takes action. Therefore, every information set $I_i \in \mathcal{I}_i$ corresponds to one decision point of player $i$ which means that $P(h_1) = P(h_2)$ and $A(h_1) = A(h_2)$ for any $h_1, h_2 \in I_i$. For convenience, we use $A(I_i)$ to represent the action set $A(h)$ and $P(I_i)$ to represent the player $P(h)$ for any $h \in I_i$. For $i \in N$, a utility function $u_i : Z \to \mathbb{R}$ specifies the payoff of player $i$ for every terminal history.

A player's behavior strategy $\sigma_i$ is a function mapping every information set of player $i$ to a probability distribution over $A(I_i)$ and $\Sigma_i$ is the set of strategies for player $i$. A strategy profile $\sigma$ is a tuple of strategies, one for each player, $(\sigma_1, \sigma_2, ..., \sigma_n)$, with $\sigma_{-i}$ referring to all the strategies in $\sigma$ except $\sigma_i$. Let $\pi^\sigma(h) = \prod_{i \in N \cup \{c\}} \pi_i^\sigma(h)$ be the probability of history $h$ occurring if all players choose actions according to $\sigma$. $\pi_i^\sigma(h)$ is the contribution of $i$ to this probability. Given a strategy profile $\sigma$, the value to player $i$ is the expected payoff of these resulting terminal nodes, $u_i(\sigma) = \sum_{z \in Z} \pi^\sigma(z) u_i(z)$.

The canonical solution concept is Nash equilibrium (NE). The strategy profile $\sigma$ forms an NE if $u_i(\sigma) \geq \max_{\sigma_i' \in \Sigma_i} u_i(\sigma_i', \sigma_{-i}), \forall i \in N$. To measure of the distance between $\sigma$ and the NE, we define $\text{NASHCONV}_i(\sigma) = \max_{\sigma_i'} u_i(\sigma_i', \sigma_{-i}) - u_i(\sigma)$ for each player and $\text{NASHCONV}(\sigma) = \sum_{i \in N} \text{NASHCONV}_i(\sigma)$. Except for the NE, some other solution concepts for extensive-form games exist conditional to different situations. For example, (Coarse) Correlated Equilibrium ((C)CE) is another popular solution concept for $n$-player general-sum games. A correlated equilibrium (CE) is a joint mixed strategy such that no player has the incentive to deviate from it. Coarse correlated equilibrium (CCE) (Moulin & Vial, 1978) is a simpler solution concept that contains CE as a subset: NE $\subseteq$ CE $\subseteq$ CCE. A strategy profile is in CCE if no player wishes to deviate before receiving a recommended signal. Similar to NE, to measure of the gap between joint strategy $\sigma$ and the (C)CE, we can use the (C)CE Gap Sum which describes how close joint policies are to (C)CE under $\sigma$ (Marris et al., 2021). In this paper, we focus not only on the NE solution but also on the CCE solution.

## 3.2 EQUILIBRIUM FINDING ALGORITHMS

**PSRO.** PSRO is initialized with a set of randomly-generated policies $\hat{\Sigma}_i$ for each player $i$. At each iteration of PSRO, a meta-game $M$ is built with all existing policies of players and then a meta-solver computes a meta-strategy, i.e., a distribution over policies of each player (e.g., Nash, $\alpha$-rank or uniform distributions). The joint meta-strategy for all players is denoted as $\boldsymbol{\alpha}$, where $\alpha_i(\sigma)$ is the probability that player $i$ takes $\sigma$ as their strategy. After that, an oracle computes at least one policy for each player, which is added to $\hat{\Sigma}_i$. We note when computing the new policy for one player, all other players' policies and the meta-strategy are fixed, which corresponds to a single-player optimization problem and can be solved by DQN (Mnih et al., 2015) or policy gradient reinforcement learning algorithms. NFSP can be seen as a special case of PSRO with uniform distributions as meta-strategies (Heinrich et al., 2015). Joint Policy Space Response Oracles (JPSRO) is a novel extension to PSRO with full mixed joint policies to enable coordination among policies (Marris et al., 2021). JPSRO is proven to converge to a (C)CE over joint policies in extensive-form games.

**CFR.** CFR is a family of iterative algorithms for approximately solving large imperfect-information games. Let $\sigma_i^t$ be the strategy used by player $i$ in round $t$. We define $u_i(\sigma, h)$ as the expected utility of player $i$ given that the history $h$ is reached, and then all players act according to strategy $\sigma$ from that point on. Let us define $u_i(\sigma, h \cdot a)$ as the expected utility of player $i$ given that the history $h$ is reached and then all players play according to strategy $\sigma$ except player $i$ who selects action $a$ in history $h$. Formally, $u_i(\sigma, h) = \sum_{z \in Z} \pi^\sigma(h, z) u_i(z)$ and $u_i(\sigma, h \cdot a) = \sum_{z \in Z} \pi^\sigma(h \cdot a, z) u_i(z)$. The *counterfactual value* $v_i^\sigma(I)$ is the expected value of an information set $I$ given that player $i$ attempts to reach it. This value is the weighted average of the value of each history in an information set. The weight is proportional to the contribution of all players other than $i$ to reach each history. Thus, $v_i^\sigma(I) = \sum_{h \in I} \pi_{-i}^\sigma(h) \sum_{z \in Z} \pi^\sigma(h, z) u_i(z)$. For any action $a \in A(I)$, the counterfactual value of action $a$ is $v_i^\sigma(I, a) = \sum_{h \in I} \pi_{-i}^\sigma(h) \sum_{z \in Z} \pi^\sigma(h \cdot a, z) u_i(z)$. The *instantaneous regret* for action $a$ in information set $I$ of iteration $t$ is $r^t(I, a) = v_{P(I)}^{\sigma^t}(I, a) - v_{P(I)}^{\sigma^t}(I)$. The *counterfactual*

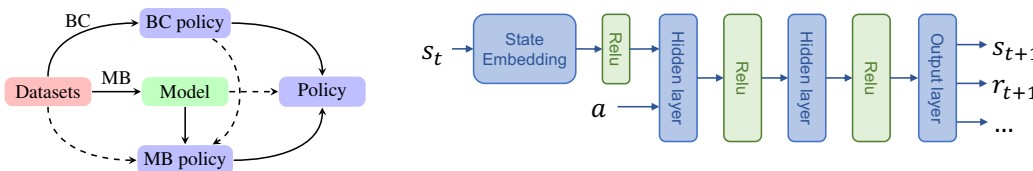

Figure 2: The flow of OEF algorithms.  Figure 3: Environment model structure.

---

**Algorithm 1:** General Framework of Offline Equilibrium Finding

---

1 Given an offline dataset $D$, train an environment model $E$;

2 Learn an equilibrium policy $\pi^{mb}$ on $E$ using any model-based OEF algorithm;

3 (Optional) Based on $D$, train a behavior policy $\pi^{bc}$ using BC technique;

4 (Optional) Combine $\pi^{bc}$ and $\pi^{mb}$ with appropriate weights $\alpha$ to get final policy $\pi$

---

*regret* for action $a$ in $I$ of iteration $T$ is $R^T(I,a) = \sum_{t=1}^{T} r^t(I,a)$. In vanilla CFR, players use *Regret Matching* to pick a distribution over actions in an information set proportional to the positive cumulative regret of those actions. Formally, in iteration $T+1$, player $i$ selects action $a \in A(I)$ according to probabilities

$$\sigma^{T+1}(I,a) = \begin{cases} \frac{R_+^T(I,a)}{\sum_{b \in A(I)} R_+^T(I,b)} & \text{if } \sum_{b \in A(I)} R_+^T(I,b) > 0, \\ \frac{1}{|A(I)|} & \text{otherwise,} \end{cases}$$

where $R_+^T(I,a) = \max\{R^T(I,a), 0\}$ because we are concerned about the cumulative regret when it is positive only. If a player acts according to regret matching in $I$ on every iteration, then in iteration $T$, $R^T(I) \le \Delta_i \sqrt{|A_i|}\sqrt{T}$ where $\Delta_i = \max_z u_i(z) - \min_z u_i(z)$ is the range of utilities of player $i$. Moreover, $R_i^T \le \sum_{I \in \mathcal{I}_i} R^T(I) \le |\mathcal{I}_i|\Delta_i\sqrt{|A_i|}\sqrt{T}$. Therefore, $\lim_{T \to \infty} \frac{R_i^T}{T} = 0$. In two-player zero-sum games, if both players' average regret $\frac{R_i^T}{T} \le \epsilon$, their average strategies $(\overline{\sigma}_1^T, \overline{\sigma}_2^T)$ form a $2\epsilon$-equilibrium (Waugh et al., 2009). Most previous works focus on tabular CFR, where counterfactual values are stored in a table. Recent works adopt deep neural networks to approximate the counterfactual values and outperform their tabular counterparts (Brown et al., 2019; Steinberger, 2019; Li et al., 2019; 2021).

## 4 ALGORITHMS FOR OFFLINE EQUILIBRIUM FINDING

In real life, offline datasets may be collected using different behavior strategies. To simulate real-world cases, we focus on four types of datasets: expert dataset, learning dataset, random dataset, and hybrid dataset. Based on the offline datasets, we give the formal definition of the offline equilibrium finding problem. Details of offline datasets, the relationship between the information set $I$ and game state $s_t$, and some theoretical analysis on the OEF problem can be found in Appendix B.

**Definition 1** (OEF). *Given a game's offline dataset $\mathcal{D} = (s_t, a, s_{t+1}, r_{t+1})$ where $s_t$ and $s_{t+1}$ refer to the game state, $a$ refer to the action played at $s_t$ and $r_{t+1}$ refer to the reward after performing action $a$ at $s_t$. The behavior strategy used to collect the dataset $\mathcal{D}$ is unknown. The OEF problem is to find an approximate equilibrium strategy profile $\sigma^*$ that achieves a small gap between $\sigma^*$ and equilibrium, i.e., the NASHCONV for NE and (C)CE Gap Sum for (C)CE, only based on $\mathcal{D}$.*

Then, we introduce our proposed model-based method (MB) for solving the OEF problem, which can adapt any online equilibrium finding algorithm into the context of the offline setting. Since the behavior cloning (BC) technique can learn a good policy when the behavior policy of the dataset is close to the equilibrium policy, which makes up for the deficiency of the model-based algorithm in this aspect, we apply the BC technique to further improve the performance of the model-based algorithm. To do this, we combine these two methods (BC+MB) by assigning different weights to trained policies using these two methods to get the final strategy (see the experiments for details).

Figure 2 and Algorithm 1 show the general procedure of the OEF algorithm. Given the offline dataset, we first train an environment model $E$ based on these game data. Then we perform an

equilibrium finding algorithm on the trained environment model to learn a model-based policy (MB policy). To further improve the performance, we first adopt the behavior cloning technique to learn a policy (BC policy) directly from the offline dataset. Then we combine these two policies, i.e., the BC policy and the MB policy, by assigning proper weights to these two policies to obtain the final policy under different cases. We leave two unexplored options for future research: i) whether we can learn an MB policy with the regularization of the BC policy, as well as interacting with the dataset, and ii) if we can use the learned model to get the proper weights when combining the two policies. Next, we move to introduce the details of our model-based method. The details of the behavior cloning technique can be found in Appendix D.

## 4.1 Environment Model

In this part, we first focus on how to train an environment model based on an OEF dataset. The ultimate goal of OEF is to learn the approximate equilibrium strategy only based on an offline dataset. However, if we directly use the data in the offline dataset to compute the equilibrium strategy, we may not be able to evaluate the game states not included in the dataset. One possibility to deal with this issue is to train an intermediate environment model using the offline dataset to capture the game information which is necessary for computing the equilibria. In other words, the purpose of the trained environment model is to learn the dynamics of the game environment and output the next game state information based on the current game state and action. Since deep neural networks have powerful generalization abilities, the neural network model may provide approximate results for game states not sufficiently captured in the data. Figure 3 shows the structure of the environment model. The environment model $E$ parameterized by $\theta_e$ takes the game state $s_t$ and action $a$ of the player, who played at the state $s_t$, as inputs and outputs the next game state $s_{t+1}$, the rewards $r_{t+1}$ of all players, and other information such as the next legal action set and whether the game ends. We use stochastic gradient descent (SGD) optimizer to perform parameter updates when training the environment model. We can employ any loss function that satisfies the conditions of Bregman divergence (Banerjee et al., 2005), such as mean squared error loss as below:

$$\mathcal{L}_{env} = \mathbb{E}_{(s_t, a, s_{t+1}, r_{t+1}) \sim D}[MSE((s_{t+1}, r_{t+1}), E(s_t, a; \theta_e))]. \tag{1}$$

## 4.2 Model-Based Algorithms

After training the environment model, we aim to compute the equilibrium strategy based on this model. To this end, we generalize existing online equilibrium finding algorithms to the context of the OEF setting by replacing the actual environment used in these online algorithms with the trained environment model. In this paper, we propose three model-based algorithms: Offline Equilibrium Finding-Policy Space Response Oracles (OEF-PSRO) and Offline Equilibrium Finding-Deep CFR (OEF-CFR) algorithms, which generalize PSRO and Deep CFR to compute NEs, and Offline Equilibrium Finding-Joint Policy Space Response Oracles (OEF-JPSRO), which generalize JPSRO to compute (C)CEs. Then we move to introduce these algorithms in detail.

In PSRO or JPSRO, a meta-game is represented as an empirical game starting with a single policy (uniform random) and iteratively enlarged by adding new policies (oracles) that approximate the best responses to the meta-strategies of other players. It is clear that when computing the best response policy oracle, interactions with the environment are required to gain game information. In the OEF setting, only an offline dataset is provided, and directly applying PSRO or JPSRO is not feasible. In OEF-PSRO and OEF-JPSRO, we use the trained environment model to replace the actual environment to provide the game information. It is a common knowledge that when computing the best response policy using DQN or other reinforcement learning algorithms, the next state and reward based on the current state and the action are required. Our environment model can provide such information. The model can also offer additional information for approximating the missing entries in the meta-game matrix using the same manner. Deep CFR is a variant of CFR which uses neural networks to approximate counterfactual regret values and average strategies. During this approximation, the game tree has to be partially traversed to arrive at the counterfactual regret value. This process requires an environment to provide the necessary game information. Akin to OEF-PSRO, also in OEF-CFR, we use the trained environment model to replace the actual environment. During the traversal, the environment needs to identify the next game state and utility for the terminal game state, for which we may employ our trained environment model. These algorithms are described in detail in Appendix D.

## 5 EXPERIMENTS

In this section, we evalutate our algorithms on the OEF datasets collected from various games. Firstly, we conduct two offline RL algorithms to verify their performance. Then, we evaluate the performance of our algorithms in computing NEs based on the OEF datasets via experiments on different offline datasets. Finally, to assess the performance of our algorithm in computing CCEs based on the OEF datasets, we conduct the OEF-JPSRO algorithm on two three-player games.

### 5.1 EXPERIMENTAL SETTING

OpenSpiel [1] is a collection of environments and algorithms for research in general reinforcement learning and search/planning in games (Lanctot et al., 2019). It is widely accepted and implements many different games. We use it as our experimental platform and opt for Kuhn poker, Leduc poker, Liar's Dice, and Phantom Tic-Tac-Toe, which are all widely used in previous works (Lisý et al., 2015; Brown & Sandholm, 2019), as experimental domains. To get the OEF datasets, we generate three datasets for every game as introduced in Appendix B and mix the random and the expert datasets in different proportions to get hybrid datasets. Then we conduct our experiments on these offline datasets. NashConv (exploitability) is adopted for measuring the strategy of how close to NEs, and (C)CE Gap Sum is used as a measurement of the closeness to (C)CEs. All the experiments are performed on a server with a 10-core 3.3GHz Intel i9-9820X CPU and an NVIDIA RTX 2080 Ti GPU. All results are averaged over three seeds, and the error bars are also reported. Only selected results are shown here. The rest is deferred to Appendix E.

### 5.2 COMPARISON WITH OFFLINE RL

In this section, we empirically show that naive offline RL algorithms are not enough for the OEF setting. To this end, we select one model-based offline RL algorithm – Model-based Offline Policy Optimization (MOPO) (Yu et al., 2020) and one model-free offline RL algorithm – Best-Action Imitation Learning (BAIL) (Chen et al., 2020) as the representative of offline RL algorithms. Figures 4(b) and 5(b) show the comparison results of two-player Kuhn poker and two-player Leduc poker games. The x-axis represents the proportion of random data in the hybrid dataset. If the ratio is zero, the dataset equals the expert dataset, and if the percentage is one, it indicates that the hybrid dataset is the random dataset. We can find that in these hybrid datasets, our algorithm performs better than these two offline RL algorithms. It also shows that the performance of the MOPO algorithm varies widely regardless of the type of the dataset. Compared to the MOPO, the performance of BAIL is somewhat related to the quality of the dataset. However, neither of these offline RL algorithms can get a strategy profile close enough to the equilibrium strategy, which may be due to the players' policies being optimized independently. These offline RL algorithms cannot perform well in the OEF setting, which verifies the claim that offline RL is not enough for the OEF setting.

### 5.3 NASH EQUILIBRIUM

Before running our OEF algorithms, we first analyze the OEF datasets. We count the frequency of leaf nodes and perform Fourier transform on it. Figures 4(a) and 5(a) show the analysis results for different datasets. It can be found that there are more high-frequency data in the expert dataset than in the random dataset. And the learning dataset and other hybrid datasets are intermediate between the expert dataset and random dataset. More analysis results of datasets can be found in Appendix C.

We first run the behavior cloning technique and model-based algorithms solely on several games based on their hybrid datasets to assess their performance. Figures 4(c) and 5(c) show the results of BC on two-player Kuhn poker and Leduc poker games. We can see that as the proportion of the random dataset increases, the performance of BC policy decreases in these two games. The finding is intuitive since the BC technique can mimic the behavior strategy in the dataset and performs well only on datasets containing more expert data. It also shows that with the increase in the size of offline data, the performance evolves more stable while the improvement is not very significant. Therefore, the performance of BC policy depends on the quality of datasets, i.e., the quality of the behavior policy used to generate the dataset. Figures 4(d) and 5(d) show the results of MB. From

---

[1]`https://github.com/deepmind/open_spiel`

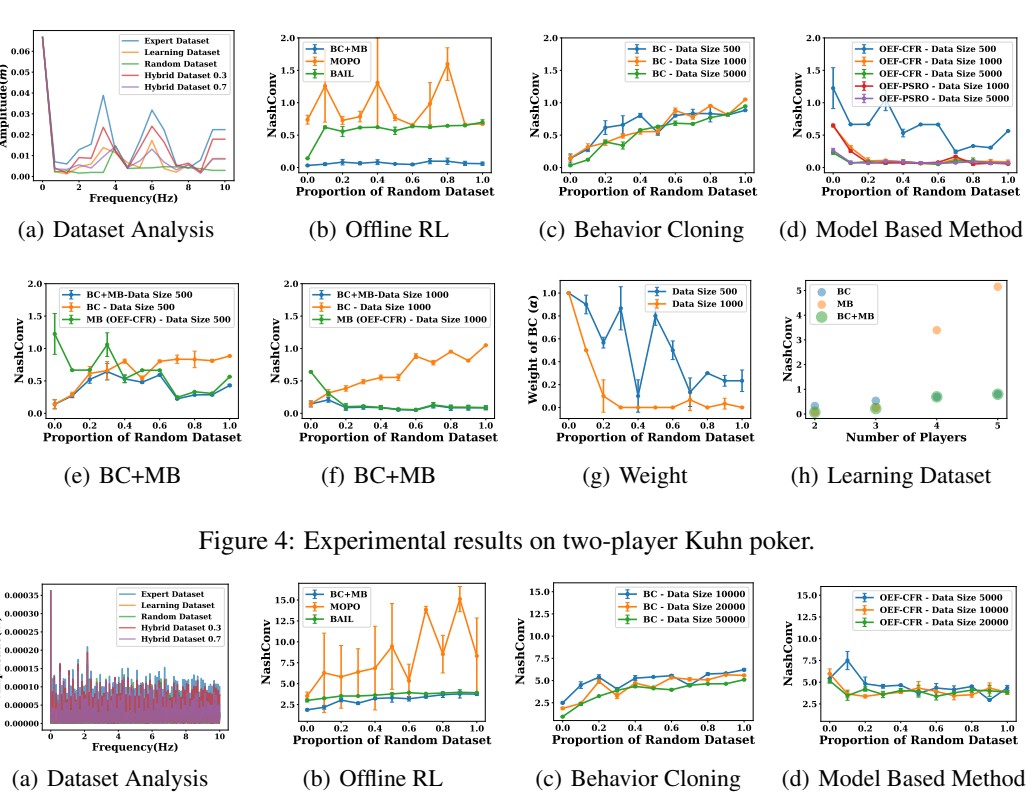

Figure 4: Experimental results on two-player Kuhn poker.

Figure 5: Experimental results on two-player Leduc poker.

Figure 4(d), we find that OEF-CFR and OEF-PSRO can get almost the same results. It indicates that the performance of MB mainly depends on the quality of the trained environment model, and we can use either algorithm to calculate MB policy. Another finding is that with the increase in the size of offline data, the performance becomes better. It indicates that if the dataset includes enough data, the trained environment model is closer to the actual environment (test environment). From the above results, we can know that the BC policy performs poorly in the random dataset and performs well in the expert dataset. However, the MB method performs slightly poorly in the expert dataset. To further improve the performance of the final strategy, a straightforward way is to merge these two policies by assigning proper weights to them in different cases.

Let us first introduce the combination method BC+MB, i.e., how to combine these two policies. Let $\alpha$ be the weight of BC policy. Then the weight of the MB policy is $1 - \alpha$. We preset 11 weight assignment plans, i.e., $\alpha \in \{0, 0.1, 0.2, ..., 0.9, 1\}$. Next, we use these 11 weight assignment plans to combine these two policies to get a set of final policies. Then we test these final policies in a real game to get the best final policy according to the exploitability. Another more complex method is to learn the weights using few-shot interactions with the real game. We left this as future work.

Figures 4(e)-4(f) and 5(e)-5(f) show the results of BC+MB on two-player Kuhn and Leduc poker games. We also plot the results of BC and MB methods for comparison. We found that the BC+MB method performs better than both BC and MB methods in all cases, which indicates that the combination is useful. The weights of BC policy ($\alpha$) which make these combined policies perform best

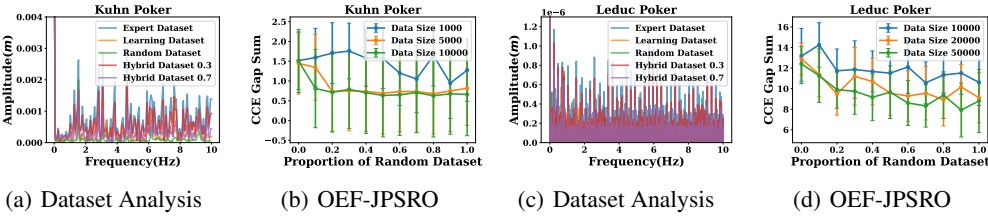

Figure 6: Experimental results on multi-player games.

on these datasets are shown in Figures 4(g) and 5(g). As the proportion of the random dataset decreases, the weight of the BC policy in the final policy increases. It adheres to the intuition that BC policy performs well when the dataset includes many expert data. In that case, the weight of the BC policy in the final policy is high. We also test the BC+MB algorithm on poker games with different players under the learning datasets which can be viewed as datasets produced by unknown strategies. Figures 4(h) and 5(h) show that BC+MB outperforms other methods in all games including multi-player Kuhn poker and Leduc poker. It indicates that given a dataset generated by an unknown strategy, the BC+MB algorithm can always get a good policy closing to the equilibrium strategy.

### 5.4 Coarse Correlated Equilibrium

To evaluate the performance of the OEF-JPSRO algorithm in computing the CCE strategy, we perform experiments on two three-player poker games under hybrid datasets. In these multi-player games, although there is no guarantee to converge to NE, we can still use PSRO with $\alpha$-rank as meta-game to get a good strategy (low exploitability) to generate the expert dataset. Here, we did not perform the behavior cloning technique since the offline dataset is collected using an independent strategy of every player instead of a joint strategy. Figure 6 shows the results on three-player Kuhn and Leduc poker games. The analysis of datasets shows the same result as before. As the size of the used offline data increases, we find that the performance of OEF-JPSRO improves. It further verifies that the performance of the model-based method mainly depends on the environment model, which may need more data to get well trained. We also run OEF-CFR algorithm on these three-player poker games (Appendix E). Since the strategy learned by OEF-CFR is not a joint strategy, we only compute the NashConv to measure how close it is to the NE.

## 6 Conclusion

We initiated an investigation of offline equilibrium finding (OEF), i.e., equilibrium finding on offline datasets. We constructed OEF datasets from four widely-used games using three data-collecting strategies. To solve the OEF problem, we proposed a model-based method that can generalize any online equilibrium finding algorithm with mere changes by introducing an environment model. Specifically, we adapted several existing online equilibrium finding algorithms to the OEF setting for computing different equilibrium solution concepts. To further improve the performance, we combined the behavior cloning technique with the model-based method. Extensive experimental results demonstrated that our proposed algorithm performs better than existing offline RL algorithms and the model-based method is necessary for the OEF setting. We hope our efforts may open new directions in equilibrium finding and accelerate the research in game theory.

**Future works.** There are several limitations of this work that we intend to tackle in the future. First, the games we considered are rather smaller and large-scale games like Texas Hold'em poker (Brown & Sandholm, 2018) were postponed till future work. Second, the types of generated offline datasets are limited. For future work, we plan to collect datasets using large-scale games and connect our library to StarCraft II Unplugged (Mathieu et al., 2021). We will also include more data-collecting strategies (e.g., bounded rational agents) as well as additional human expert data[2] to diversify the provided datasets. Moreover, we will investigate the relations between data distributions and OEF algorithms to characterize the influence of the data on the performances of OEF algorithms.

---

[2]http://poker.cs.ualberta.ca/irc_poker_database.html

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

## A  RELATED WORK OVERVIEW

**Offline Reinforcement Learning (Offline RL).** Offline RL is a *data-driven* paradigm that learns exclusively from static datasets of previously collected interactions, making it feasible to extract policies from large and diverse training datasets (Levine et al., 2020). This paradigm can be extremely valuable in settings where online interaction is impractical, either because data collection is expensive or dangerous (e.g., in robotics (Singh et al., 2021), education (Singla et al., 2021), healthcare (Liu et al., 2020), and autonomous driving (Kiran et al., 2022)). Therefore, efficient offline RL algorithms have a much broader range of applications than online RL and are particularly appealing for real-world applications (Prudencio et al., 2022). Due to its attractive characteristics, there have been a lot of recent studies. Here, we can divide the research of Offline RL into two categories: model-based and model-free algorithms.

Model-free algorithms mainly use the offline dataset directly to learn a good policy. When learning the strategy from an offline dataset, we have two types of algorithms: actor-critic and imitation learning methods. Those actor-critic algorithms focus on implementing policy regularization and value regularization based on existing reinforcement learning algorithms. Haarnoja et al. (Haarnoja et al., 2018) propose soft actor-critic (SAC) by adding an entropy regularization term to the policy gradient objective. This work mainly focuses on policy regularization. For the research of value regularization, an offline RL method named Constrained Q-Learning (CQL) (Kumar et al., 2020) learns a lower bound of the true Q-function by adding value regularization terms to its objective. Another line of research on learning a policy is imitation learning which mimics the behavior policy based on the offline dataset. Chen et al. (Chen et al., 2020) propose a method named Best-Action Imitation Learning (BAIL), which fits a value function, then uses it to select the best actions. Meanwhile, Siegel et al. (Siegel et al., 2020) propose a method that learns an Advantage-weighted Behavior Model (ABM) and uses it as a prior in performing Maximum a-posteriori Policy Optimization (MPO) (Abdolmaleki et al., 2018). It consists of multiple iterations of policy evaluation and prior learning until they finally perform a policy improvement step using their learned prior to extracting the best possible policy. Model-based algorithms rely on the offline dataset to learn a dynamics model or a trajectory distribution used for planning. We use the trajectory distribution induced by models to determine the best set of actions to take at each given time step. Kidambi et al. (Kidambi et al., 2020) propose a method named Model-based Offline Reinforcement Learning (MOReL), which measures their model's epistemic uncertainty through an ensemble of dynamics models. Meanwhile, Yu et al. (Yu et al., 2020) propose another method named Model-based Offline Policy Optimization (MOPO), which uses the maximum prediction uncertainty from an ensemble of models. Concurrently, Matsushima et al. (Matsushima et al., 2020) propose the BehaviorREgularized Model-ENsemble (BREMEN) method, which learns an ensemble of models of the behavior MDP, as opposed to a pessimistic MDP. In addition, it implicitly constrains the policy to be close to the behavior policy through trust-region policy updates. More recently, Yu et al. (Yu et al., 2021a) proposed a method named Conservative Offline Model-Based policy Optimization (COMBO), a model-based version of CQL. The main advantage of COMBO concerning MOReL and MOPO is that it removes the need for uncertainty quantification in model-based offline RL approaches, which is challenging and often unreliable. In the OEF setting, these above Offline RL algorithms can not directly apply to it, which we have described in section 2 and experimental results empirically verify this claim.

**Empirical Game Theoretic Analysis (EGTA).** Empirical Game Theoretic Analysis is an empirical methodology that bridges the gap between game theory and simulation for practical strategic reasoning (Wellman, 2006). In EGTA, game models are iteratively extended through a process of generating new strategies based on learning from experience with prior strategies. The strategy exploration problem (Jordan et al., 2010) that how to efficiently assemble an efficient portfolio of policies for EGTA is the most challenging problem in EGTA.

Schvartzman & Wellman (Schvartzman & Wellman, 2009b) deploy tabular RL as a best-response oracle in EGTA for strategy generation. They also build the general problem of strategy exploration in EGTA and investigate whether better options exist beyond best-responding to an equilibrium (Schvartzman & Wellman, 2009a). Investigation of strategy exploration was advanced significantly by the introduction of the Policy Space Response Oracle (PSRO) framework (Lanctot et al., 2017) which is a flexible framework for iterative EGTA, where at each iteration, new strategies are generated through reinforcement learning. Note that when employing NE as the meta-strategy

solver, PSRO reduces to the double oracle (DO) algorithm (McMahan et al., 2003). In the OEF setting, only an offline dataset is provided, and there is no accurate simulator. In EGTA, a space of strategies is examined through simulation, which means that it needs a simulator, and the policies are known in advance. Therefore, techniques in EGTA cannot directly apply to OEF.

**Opponent Modeling (OM) in Multi-Agent Learning.** Opponent modeling algorithm is necessary in multi-agent settings where secondary agents with competing goals also adapt their strategies, yet it remains challenging because policies interact with each other and change (He et al., 2016). One simple idea of opponent modeling is to build a model each time a new opponent or group of opponents is encountered (Zheng et al., 2018). However, it is infeasible to learn a model every time. A better approach is to represent an opponent's policy with an embedding vector. Grover et al. (Grover et al., 2018) use a neural network as an encoder, taking the trajectory of one agent as input. Imitation learning and contrastive learning are also used to train the encoder. Then, the learned encoder can be combined with RL by feeding the generated representation into the policy or/and value network. DRON (He et al., 2016) and DPIQN (Hong et al., 2017) are two algorithms based on DQN, which use a secondary network that takes observations as input and predicts opponents' actions. However, if the opponents can also learn, these methods become unstable. So it is necessary to take the learning process of opponents into account.

Foerster et al. (Foerster et al., 2017) propose a method named Learning with Opponent-Learning Awareness (LOLA), in which each agent shapes the anticipated learning of the other agents in the environment. Further, the opponents may still be learning continuously during execution. Therefore, Al-Shedivat et al. (Al-Shedivat et al., 2017) propose a method based on a meta policy gradient named Mata-MPG. It uses trajectories from current opponents to perform multiple meta-gradient steps and constructs a policy that favors updating the opponents. Meta-MAPG (Kim et al., 2021) extends this method by including an additional term that accounts for the impact of the agent's current policy on the future policies of opponents, similar to LOLA. Yu et al. (Yu et al., 2021b) propose model-based opponent modeling (MBOM), which employs the environment model to adapt to various opponents. In the OEF setting, our goal is to compute the equilibrium strategy based on the offline dataset. Applying opponent modeling is not enough for calculating the equilibrium strategy in the OEF setting since the opponent will always best respond to the agent.

**Equilibrium Finding Algorithms.** The contemporary state-of-the-art algorithms for solving imperfect-information extensive-form games may be roughly divided into two groups: no-regret methods derived from CFR, and incremental strategy-space generation methods of the PSRO framework. For the first group, some variants are proposed to solve large-scale imperfect-information extensive-form games. Some sampling-based CFR variants (Lanctot et al., 2009; Gibson et al., 2012; Schmid et al., 2019) are proposed to effectively solve large-scale games by traversing a subset of the game tree instead of the whole game tree. With the development of deep learning techniques, neural network function approximation is also applied to CFR algorithm. Deep CFR (Brown et al., 2019), Single Deep CFR (Steinberger, 2019) and Double Neural CFR (Li et al., 2019) are algorithms using deep neural networks to replace the tabular representation in the CFR algorithm. For the second group, PSRO (Lanctot et al., 2017) is a general framework that scales Double Oracle (DO) (McMahan et al., 2003) to large extensive-form games via using reinforcement learning to compute the best response strategy approximately. To make PSRO more effective in solving large-scale games, Pipeline PSRO (P2SRO) (McAleer et al., 2020) is proposed by parallelizing PSRO with convergence guarantees. Extensive-Form Double Oracle (XDO) (McAleer et al., 2021) is a version of PSRO where the restricted game allows mixing population strategies not only at the root of the game but every information set. It can guarantee to converge to an approximate NE in a number of iterations that are linear in the number of information sets, while PSRO may require a number of iterations exponential in the number of information sets. Neural XDO (NXDO) as a neural version of XDO learns approximate best response strategies through any deep reinforcement learning algorithm. Recently, Anytime Double Oracle (ADO) (McAleer et al., 2022), a tabular double oracle algorithm for 2-player zero-sum games is proposed to converge to a Nash equilibrium while decreasing exploitability from one iteration to the next. Anytime PSRO (APSRO) as a version of ADO calculates best responses via reinforcement learning algorithms. Except for NEs, other equilibrium solution concepts, for example, (Coarse) Correlated equilibrium ((C)CE) are considered. Joint Policy Space Response Oracles (JPSRO) (Marris et al., 2021) is proposed for training agents in n-player, general-sum extensive-form games, which provably converges to (C)CEs. The excellent performance of these equilibrium finding algorithms depends on the existence of efficient and accu-

rate simulators. However, constructing a sufficiently accurate simulator may not be feasible or very expensive. In this case, we may resort to offline equilibrium finding (OEF) where the equilibrium strategy is computed based on the previous game data.

# B   DATASET

In this section, we describe four types of data sets and how to collect them.

**Random dataset.** This type of dataset is collected using a random strategy. The motivation behind this is that when we do not know how to play a game, we may try to explore it randomly. The process of collecting is akin to that of a beginner familiarizing themselves with the game for the first time. The dataset is collected in three steps. First, we assume that every player plays a uniform strategy. Second, the players engage in the game repeatedly. Finally, we collect the generated game data. Note that in imperfect-information extensive-form games, the definition of information sets corresponds to the decision points of players. Here, we take the information sets of all players in the game as a game state. Therefore, the game data include the transition information such as current game state $s_t = (I_1^t, I_2^t, ..., I_n^t)$, action $a$, next game state $s_{t+1} = (I_1^{t+1}, I_2^{t+1}, ..., I_n^{t+1})$, and other game information, e.g., whether the game ends and players' rewards $r_{t+1} = (r_1^{t+1}, r_2^{t+1}, ..., r_n^{t+1})$.

**Learning dataset.** This type of dataset is collected when learning the Nash equilibrium strategy. When solving a game using some existing equilibrium finding algorithm, the players have to interact with the game environment. During the course of learning, we may gather these intermediate interaction game data and store them as a learning dataset. In contrast to the random dataset, the players' strategies gradually improve as we get closer to Nash equilibria.

**Expert dataset.** This type of dataset is collected using an NE strategy. The motivation behind the dataset is that when learning a game, we often prefer to observe more experienced players at play. We simulate the expert players using the NE strategy and collect the interaction data. We follow a similar methodology as with the random dataset. First, we compute the NE strategies using any existing equilibrium finding algorithm. As a second step, the Nashian players repeatedly interact in the game. Finally, we gather the generated data and store them as the expert dataset. In multiplayer or general-sum games, although CFR or PSRO cannot converge to NE, we also apply these algorithm for collecting the expert dataset. Although there is no guarantee, we can still get a good strategy using these algorithms, for example, PSRO with $\alpha$-rank as the meta-solver Muller et al. (2019) can get pretty good strategy (low exploitability) under general-sum many-player games.

**Hybrid dataset.** In addition to the three types of datasets mentioned above, we also consider hybrid datasets consisting of random and expert interactions mixed in different ratios to simulate offline datasets generated by unknown strategies.

We collect the data from player engagements in the most frequently used benchmarking imperfect-information extensive-form games in contemporary research on the equilibrium finding. These games include poker games (two-player and multi-player Kuhn poker, two-player and multi-player Leduc poker), Phantom Tic-Tac-Toe, and Liar's Dice.

## B.1   INFLUENCE OF DATASET COVERAGE

The concurrent work Cui & Du (2022) investigates the necessary properties of offline datasets of two-player zero-sum Markov games to successfully infer their NEs. To do this, they proposed several dataset coverage assumptions. Following the work by Cui & Du (2022), we can also define two assumptions on the dataset coverage of offline datasets. These theorem results are all for computing Nash equilibria in two-player zero-sum games.

**Assumption 1.** *The NE strategy $\sigma^*$ is covered by the expert dataset.*

**Assumption 2.** *For $\forall s_t, a \in A(s_t)$, $(s_t, a, s_{t+1})$ is covered by the random dataset.*

From the empirical analysis, we know that the performance of the model-based algorithm mainly depends on the gap between the trained environment model and the actual game environment. It means that if the trained environment model can recover all the dynamics of the actual game, then the performance is good. Otherwise, the performance is worse. Then we have the following theorem.

**Theorem 1.** *Assuming that the environment model is well-trained on the offline dataset, the OEF-CFR/OEF-PSRO can converge to equilibrium strategy under the random dataset satisfying assumption 2 and cannot guarantee to converge under the expert dataset satisfying assumption 1.*

*Proof.* Since the environment model is well-trained on the offline dataset, the environment model can fully represent the information of the offline dataset. If the random dataset is the offline dataset, the game defined by the trained environment model is the same as the actual game. The reason is that every state transition is covered by the random dataset according to assumption 2. Then the strategy learned by OEF-CFR/OEF-PSRO is the approximate equilibrium strategy of the actual game due to the convergence property of CFR/PSRO. Therefore, the OEF-CFR/OEF-PSRO can converge to an equilibrium strategy under the random dataset satisfying assumption 2.

If the offline dataset is the expert dataset, then the dataset only covers these state transitions related to the NE strategy according to assumption 1. Therefore, the state transition of the actual game may not be covered by the expert dataset. The environment model trained based on the expert dataset would produce different transition information on these states not shown in the dataset compared with the actual game. It would cause a gap between the trained environment model and the actual game. Although OEF-CFR/OEF-PSRO can learn an approximate equilibrium strategy of the game defined by the environment model, there is no guarantee that the learned strategy is the equilibrium strategy of the actual game. □

Theorem 1 is consistent with the conclusion in Cui & Du (2022) that dataset coverage satisfying assumption 1 is insufficient for NE identification, and dataset coverage satisfying assumption 2 is sufficient for NE identification. And our empirical results also verify this conclusion. The OEF-CFR/OEF-PSRO performs best under the random dataset and worst under the expert dataset.

In this paper, to offset the drawback of the model-based algorithm under the expert dataset, we propose to combine the behavior cloning technique. From the introduction of the behavior cloning technique, we know that BC can mimic the behavior policy in the dataset. Therefore, we have the following theorem describing the power of the BC technique.

**Theorem 2.** *Assuming that the behavior cloning policy is well-trained on the offline dataset, the behavior cloning technique can get the equilibrium strategy under the expert dataset satisfying assumption 1, and cannot get the equilibrium strategy under the random dataset satisfying assumption 2.*

*Proof.* The assumption that the behavior cloning policy is well-trained on the offline dataset means that the behavior cloning policy can precisely mimic the behavior strategy used to generate the offline dataset. If the offline dataset is the expert dataset, according to assumption 1, the behavior strategy of the dataset is the NE strategy. Therefore, applying the behavior cloning algorithm on the expert dataset can get an NE strategy.

If the offline dataset is the random dataset, according to the generation process of the random dataset and assumption 2, the behavior strategy of the random dataset is the random strategy. Therefore, the behavior cloning algorithm can only get a random strategy instead of the equilibrium strategy under the random dataset. □

Our experimental results also show the same outcomes as Theorem 2. The performance of the behavior cloning technique mainly depends on the quality of the behavior strategy. The behavior cloning technique only performs well under the expert dataset. Based on the above two theorems, we propose to combine two techniques, the model-based algorithm and behavior cloning algorithm, to get better results under these datasets with unknown behavior strategies.

**Theorem 3.** *Under the assumptions in Theorem 1 and 2, there exists a parameter $\alpha$ such that BC+MB can get the equilibrium strategy under either the random dataset satisfying assumption 2 or the expert dataset satisfying assumption 1.*

*Proof.* In the BC+MB algorithm, the weight of the BC policy is represented by $\alpha$. The weight of the MB policy is $1-\alpha$. The $\alpha$ ranges from 0 to 1. When under the random dataset satisfying assumption 2, let $\alpha$ equal 0. Then the policy of BC+MB would equal to MB policy, i.e., the policy trained using

the model-based algorithm. According to Theorem 1, the model-based algorithm (OEF-CFR/OEF-PSRO) can converge to an equilibrium strategy under the random dataset satisfying assumption 2. Therefore, BC+MB can also converge to an equilibrium strategy under the random dataset satisfying assumption 2.

When under the expert dataset satisfying assumption 1, let $\alpha$ equal to 1. Then the policy of BC+MB would be equal to BC policy, i.e., the policy trained by behavior cloning. Similarly, according to Theorem 2, BC+MB can get an equilibrium strategy in the expert dataset satisfying assumption 1. □

Let's move to a more general case in which the offline dataset is generated by a behavior strategy $\sigma$. Then we have the following theorems under the general case.

**Theorem 4.** *Assuming that the offline dataset $\mathcal{D}_\sigma$ generated by the behavior strategy $\sigma$ covers $(s_t, a, s_{t+1}), \forall s_t, a \in A(s_t)$ and the environment model is well-trained on $\mathcal{D}_\sigma$, the model-based algorithm (OEF-CFR/OEF-PSRO) can converge to an equilibrium strategy that performs equal even better than $\sigma$.*

*Proof.* According to the proof of Theorem 1, since every state transition of the actual game is covered by $\mathcal{D}_\sigma$, the trained environment model would be the same as the actual game under the assumption that the environment model is well-trained on the offline dataset. Then according to Theorem 1, the model-based algorithm can converge to an equilibrium strategy. If $sigma$ used to generate the dataset is not the equilibrium strategy, then the model-based algorithm can get a better strategy (equilibrium strategy) than $\sigma$. And if $\sigma$ is an equilibrium strategy, then the strategy trained using a model-based algorithm would perform equal to $\sigma$. □

**Theorem 5.** *Assuming that the behavior cloning policy is well-trained on the offline dataset $\mathcal{D}_\sigma$ generated by the behavior strategy $\sigma$, the performance of behavior cloning policy $\sigma^{bc}$ would be as good as the performance of $\sigma$.*

*Proof.* According to the assumption, behavior cloning can precisely mimic the behavior strategy in the offline dataset. Therefore, $\sigma^{bc}$ would be same as $\sigma$. Consequently, the performance of $\sigma^{bc}$ would have the same performance as $\sigma$. □

**Theorem 6.** *Assuming that the environment model and the behavior cloning policy are well-trained, under the offline dataset $\mathcal{D}_\sigma$ generated using $\sigma$, BC+MB can get the strategy which is at least as good as $\sigma$. If the offline dataset $\mathcal{D}_\sigma$ covers $(s_t, a, s_{t+1}), \forall s_t, a \in A(s_t)$, BC+MB can get an equal or better strategy than $\sigma$.*

*Proof.* Following the proof of Theorem 3, let $\alpha$ equal 1. Then BC+MB would reduce to BC. Then according to Theorem 5, the performance of BC policy is at least as good as $\sigma$. Therefore, BC+MB can get a strategy that is at least as good as $\sigma$.

In another extreme case in which $\mathcal{D}_\sigma$ covers $(s_t, a, s_{t+1}), \forall s_t, a \in A(s_t)$, let $\alpha$ equal to 0. Then BC+MB would reduce to MB. Then according to Theorem 4, the MB policy performs equal to or better than $\sigma$. Therefore, in this case, BC+MB can get an equal or better strategy than $\sigma$. □

In conclusion, under the above assumptions, BC+MB can perform at least equal to the behavior strategy used to generate the offline dataset. The improvement over the behavior strategy mainly depends on the performance of the model-based algorithm under the offline dataset.

## C  VISUALIZATION OF DATASETS

In this section, we describe the visualization methods of the datasets. First, we plot the game tree. Figure 7 shows an example of the game tree. Then, we traverse the game tree using depth-first search (DFS) and index each leaf node according to the DFS results. Finally, we count the frequency of the leaf node in each dataset.

Figure 8 show the frequency of leaf node in datasets. We can find that in the random dataset, the frequency of leaf nodes is almost uniform, while in the expert dataset, the frequency distribution of

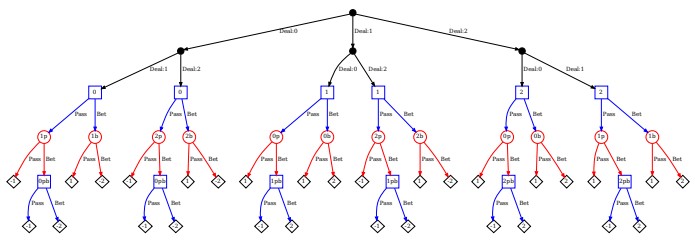

Figure 7: 2-player Kuhn poker

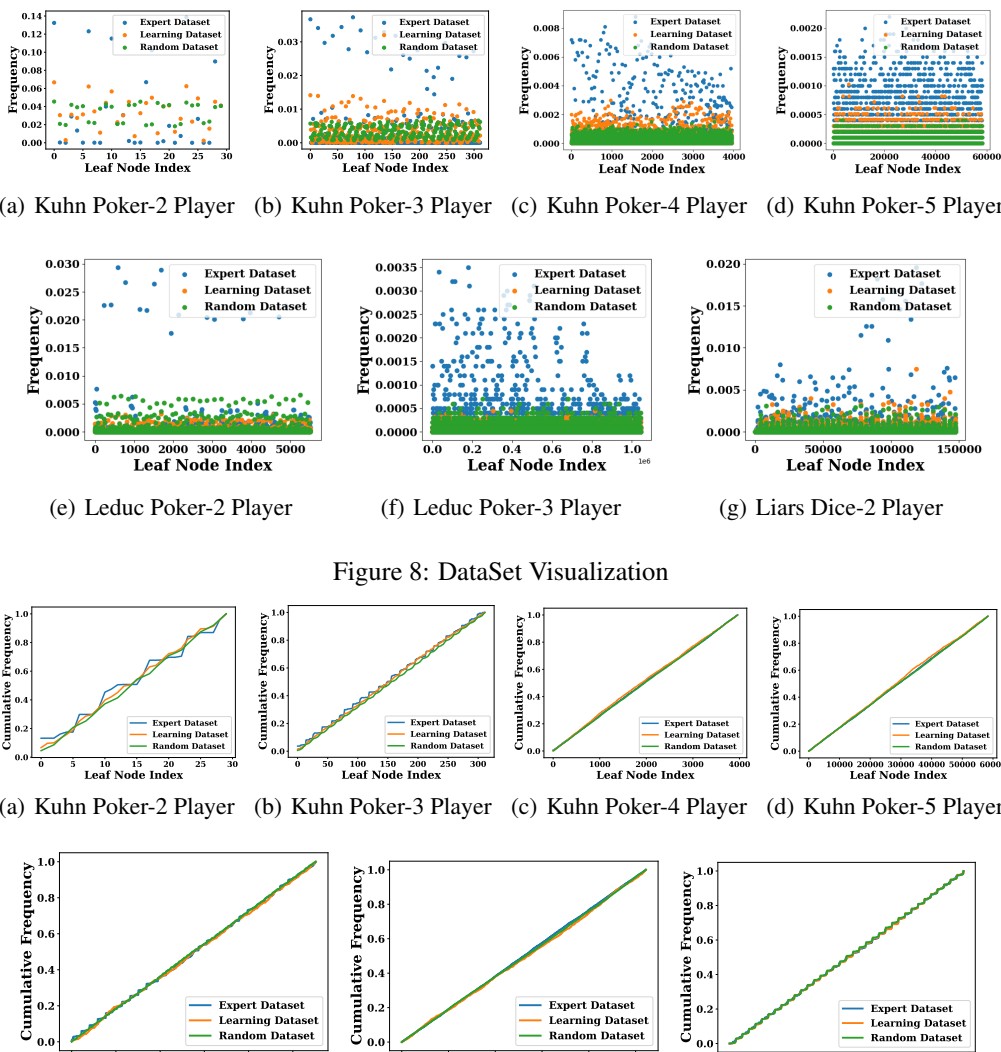

(a) Kuhn Poker-2 Player  (b) Kuhn Poker-3 Player  (c) Kuhn Poker-4 Player  (d) Kuhn Poker-5 Player

(e) Leduc Poker-2 Player  (f) Leduc Poker-3 Player  (g) Liars Dice-2 Player

Figure 8: DataSet Visualization

(a) Kuhn Poker-2 Player  (b) Kuhn Poker-3 Player  (c) Kuhn Poker-4 Player  (d) Kuhn Poker-5 Player

(e) Leduc Poker-2 Player  (f) Leduc Poker-3 Player  (g) Liars Dice-2 Player

Figure 9: DataSet Visualization

leaf nodes is uneven. The distribution of the learning dataset is between the expert dataset and the random dataset. Figure 9 shows the cumulative frequency of leaf nodes. These figures also exhibit the same result.

# D   IMPLEMENTATION DETAILS

**Behavior Cloning.** Behavior cloning (BC) is a method that mimics the behavior policy in the dataset. Behavior cloning technique is used frequently in Offline RL (Fujimoto & Gu, 2021). In the OEF setting, we can also use the BC technique to learn a behavior cloning strategy of every player from the offline dataset. More specifically, we can use the imitation learning algorithm to train a policy network $\sigma_i$ parameterized by $\theta$ for every player $i$ to predict the strategy given any information set $I_i$. Only the information sets and actions data are needed when training the behavior cloning strategy. We employ the cross-entropy loss as the training loss, defined as

$$\mathcal{L}_{bc} = \mathbb{E}_{(I_i,a)\sim D}[l(a, \sigma_i(I_i;\theta))] = \mathbb{E}_{(I_i,a)\sim D}[-\sum_{i=1}^{K} a \cdot \log(\sigma_i(I_i;\theta))], \qquad (2)$$

where $K$ is the number of actions. Figure 10 shows the structure of the behavior cloning policy network. Because equilibrial strategies in most information sets are non-trivial probability distributions, we apply a softmax layer after the output layer to obtain the final mixed strategy.

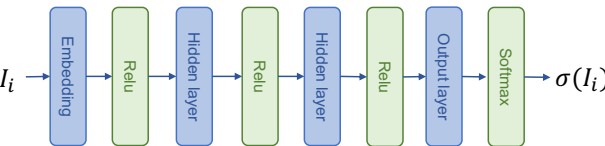

Figure 10: Structure of Behavior Cloning policy network.

**Model-based Method.** Next, we introduce our offline model-based algorithms: OEF-PSRO and OEF-CFR, which are adaptions from two widely-used online equilibrium finding algorithms PSRO and Deep CFR, and OEF-JPSRO, which is an adaption from JPSRO. These three algorithms perform on the well-trained environment model $E$. We first introduce the OEF-PSRO algorithm, and the whole flow is shown in Algorithm 2. Firstly, we need the well-trained environment model $E$ as input and initialize policy sets $\Pi$ for all players using random strategies. Then, we need to estimate a meta-game matrix by computing expected utilities for each joint strategy profile $\pi \in \Pi$. In vanilla PSRO, to get the expected utility for $\pi$, it needs to perform the strategy $\pi$ in the actual game simulator. However, the simulator is missing in the OEF setting. Therefore, we use the well-trained environment model $E$ to replace the game simulator to provide the information needed in the algorithm. Then we initialize meta-strategies using a uniform strategy. Next, we need to compute the best response policy oracle for every player and add the best response policy oracle to their policy sets. When training the best response policy oracle using DQN or other reinforcement learning algorithms, we sample the training data based on the environment model $E$. After that, we compute missing entries in the meta-game matrix and calculate meta-strategies for the meta-game. To calculate the meta-strategy $\sigma$ of the meta-game matrix, we can use the Nash solver or $\alpha$-rank algorithm. Here, we use the $\alpha$-rank algorithm as the meta solver because our algorithm needs to solve multi-player games. Finally, we repeat the above process until satisfying the convergence conditions. Since the process of JPSRO is similar to PSRO except for the best response computation and meta distribution solver, OEF-JPSRO is also similar to OEF-PSRO. We do not cover OEF-JPSRO in detail here.

Algorithm 3 shows the process of OEF-CFR. It also needs the well-trained environment model $E$ as input. We first initialize regret and strategy networks for every player and then initialize regret and strategy memories for every player. Then we need to update the regret network for every player. To do this, we can perform the traverse function to collect corresponding training data. The traverse function can be any sampling-based CFR algorithm. Here, we use the external sampling algorithm. Note that we need to perform the traverse function on the game tree. In OEF-CFR, the trained environment model can replace the game tree. Therefore, the trained environment model is the input of the traverse function. Algorithm 4 shows the process of the traverse function. In this traverse function, we collect the regret training data of the traveler, and the strategy training data of other players are also gathered. After performing the traverse function several times, the regret network is updated using the regret memory. We need to repeat the above processes $n$ iterations. Then the average strategy network for every player is trained based on its corresponding strategy memory. Finally, the trained average strategy networks are output as the approximate NE strategy.

---

**Algorithm 2:** Offline Equilibrium Finding - Policy-Space Response Oracles

---

**Input:** Trained environment model $E$;

1 Initial policy sets $\Pi$ for all players;
2 Compute expected utilities $U^{\Pi}$ for each joint $\pi \in \Pi$ **based on the environment model** $E$;
3 Initialize mate-strategies $\sigma_i = \text{UNIFORM}(\Pi_i)$ ;
4 **repeat**
5     **for** *player* $i \in [1, .., n]$ **do**
6         **for** *best response episodes* $p \in [0, ..., t]$ **do**
7             Sample $\pi_{-i} \sim \sigma_{-i}$;
8             Train best response oracle $\pi_i'$ over $\rho \sim (\pi_i', \pi_{-i})$, which **samples on the**
               **environment model** $E$;
9         **end**
10         $\Pi_i = \Pi_i \cup \{\pi_i'\}$;
11     **end**
12     Compute missing entries in $U^{\Pi}$ from $\Pi$ **based on the environment model** $E$;
13     Compute a meta-strategy $\sigma$ from $U^{\Pi}$ using $\alpha$-rank algorithm;
14 **until** *Meet convergence condition*;

**Output:** current solution strategy $\sigma_i$ for player $i$

---

**Algorithm 3:** Offline Equilibrium Finding-Deep Counterfactual Regret Minimization

---

**Input:** Trained environment model $E$;

1 Initialize regret network $R(I, a|\theta_{r,p})$ for every player $p$;
2 Initialize average strategy network $S(I, a|\theta_{\pi,p})$ for every player $p$;
3 Initialize regret memory $M_{r,p}$ and strategy memory $M_{\pi,p}$ for every player $p$;
4 **for** *CFR Iteration* $t = 1$ *to* $T$ **do**
5     **for** *player* $p \in [1, ..., n]$ **do**
6         **for** *traverse episodes* $k \in [1, ..., K]$ **do**
7             TRVERSE$(\phi, p, \theta_{r,p}, \theta_{\pi,-p}, M_{r,p}, M_{\pi,-p}, E)$;
8             # use sample algorithm to traverse game tree and record regret and strategy
9         **end**
10         Train $\theta_{r,p}$ from scratch based on regret memory $M_{r,p}$;
11     **end**
12 **end**
13 Train $\theta_{\pi,p}$ based on strategy memory $M_{\pi,p}$ for every player $p$;

**Output:** $\theta_{\pi,p}$ for every player $p$

---

# E  ADDITIONAL EXPERIMENTAL RESULTS

In this part, we provide more experimental results. First, we provide the experimental results of the behavior cloning method and model-based method (OEF-CFR) based on hybrid datasets, and then the results of our OEF algorithm (BC+MC) are given. We also test the BC+MB on two-player Phantom Tic-Tac-Toe game using the learning dataset. Finally, we provide the setting of hyper-parameters used in our experiments.

Figure 11 shows the results of the behavior cloning technique on several multi-player poker games and one two-player Liar's Dice game. It shows that as the proportion of random datasets increases, the performance decreases. It is consistent with the results of previous experiments. Therefore, the behavior cloning technique is inefficient for solving the OEF problem since it can only perform well in the expert dataset.

The experimental results of the model-based algorithm (OEF-CFR) on these games are shown in Figure 12. Since the strategy learned by OEF-CFR is not a joint strategy, we only use NashConv to measure how it is close to NEs. We found that the performance of the MB method is not stable in these games but still shows a slight decrease tendency with the increase of the proportion of the random dataset. Since the performance of the MB method mainly depends on the trained environ-

---

**Algorithm 4:** TRVERSE$(s, p, \theta_{r,p}, \theta_{\pi,-p}, M_{r,p}, M_{\pi,-p}, E)$-External Sampling Algorithm

---

1  **if** *s is terminal state* **then**
2  │ Get the utility $u_i(s)$ from environment model $E$;
3  │ **return** $u_i(s)$
4  **else if** *s is a chance state* **then**
5  │ Sample an action $a$ based on the probability $\sigma_c(s)$, which is obtained from model $E$;
6  │ $s' = E(s, a)$;
7  │ **return** TRAVERSE$(s', p, \theta_{r,p}, \theta_{\pi,-p}, M_{r,p}, M_{\pi,-p}, E)$
8  **else if** $P(s) = p$ **then**
9  │ $I \leftarrow s[p]$; # game state is formed by information sets of every player ;
10  │ $\sigma(I) \leftarrow$ strategy of $I$ computed using regret values $R(I, a|\theta_{r,p})$ based on regret matching;
11  │ **for** $a \in A(s)$ **do**
12  │ │ $s' = E(s, a)$;
13  │ │ $u(a) \leftarrow$ TRAVERSE$(s', p, \theta_{r,p}, \theta_{\pi,-p}, M_{r,p}, M_{\pi,-p}, E)$;
14  │ **end**
15  │ $u_\sigma \leftarrow \sum_{a \in A(s)} \sigma(I, a) u(a)$;
16  │ **for** $a \in A(s)$ **do**
17  │ │ $r(I, a) \leftarrow u(a) - u_\sigma$;
18  │ **end**
19  │ Insert the infoset and its action regret values $(I, r(I))$ into regret memory $M_{r,p}$;
20  │ **return** $u_\sigma$
21  **else**
22  │ $I \leftarrow s[p]$;
23  │ $\sigma(s) \leftarrow$ strategy of $I$ computed using regret value $R(I, a|\theta_{r,-p})$ based on regret matching;
24  │ Insert the infoset and its strategy $(I, \sigma(s))$ into strategy memory $M_{\pi,-p}$;
25  │ Sample an action $a$ from the probability distribution $\sigma(s)$;
26  │ $s' = E(s, a)$;
27  │ **return** TRAVERSE$(s', p, \theta_{r,p}, \theta_{\pi,-p}, M_{r,p}, M_{\pi,-p}, E)$;
28  **end**

---

ment model, it indicates that in these games, the trained environment model is far from the actual game environment which is also the test game environment. Therefore, learning a good enough environment model is a big challenge in these games with more uncertain factors.

Figure 13(a)-13(j) show the experimental results of BC+MB on various games. We also test our OEF algorithm BC+MB in Phantom Tic-Tac-Toe game based on the learning dataset (Figure 13(k)). The NashConv values in Phantom Tic-Tac-Toe are approximate results since the best response policy is trained using DQN, and the utilities are obtained by simulation. It shows that the BC+MB performs better than BC and MB, which implies that our combination method can perform well in any game under any unknown dataset. The proper weights in the BC+MB algorithm under different datasets are shown in Figure 14.

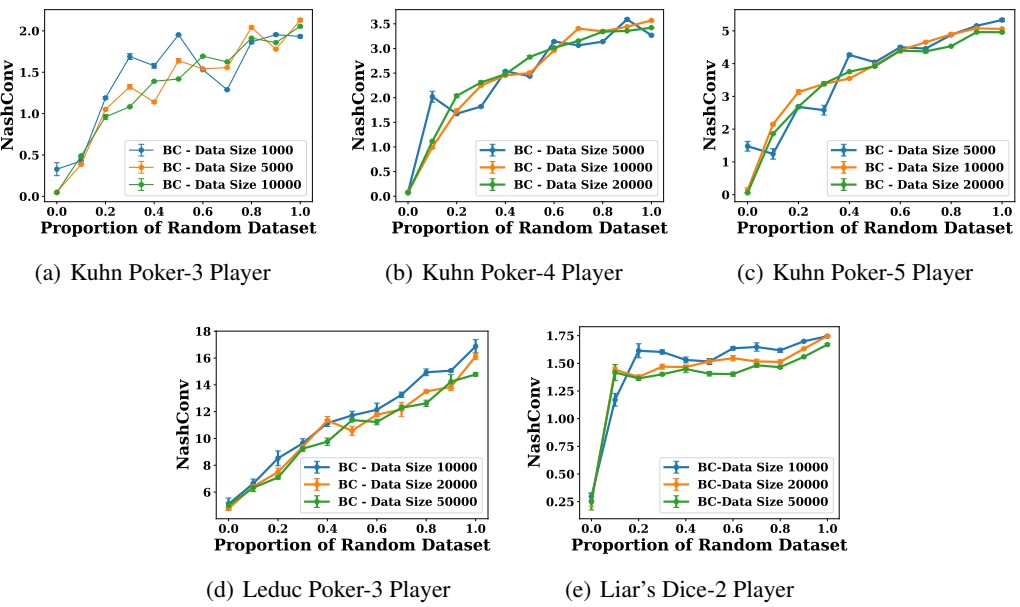

(a) Kuhn Poker-3 Player

(b) Kuhn Poker-4 Player

(c) Kuhn Poker-5 Player

(d) Leduc Poker-3 Player

(e) Liar's Dice-2 Player

Figure 11: Experimental results for the BC method

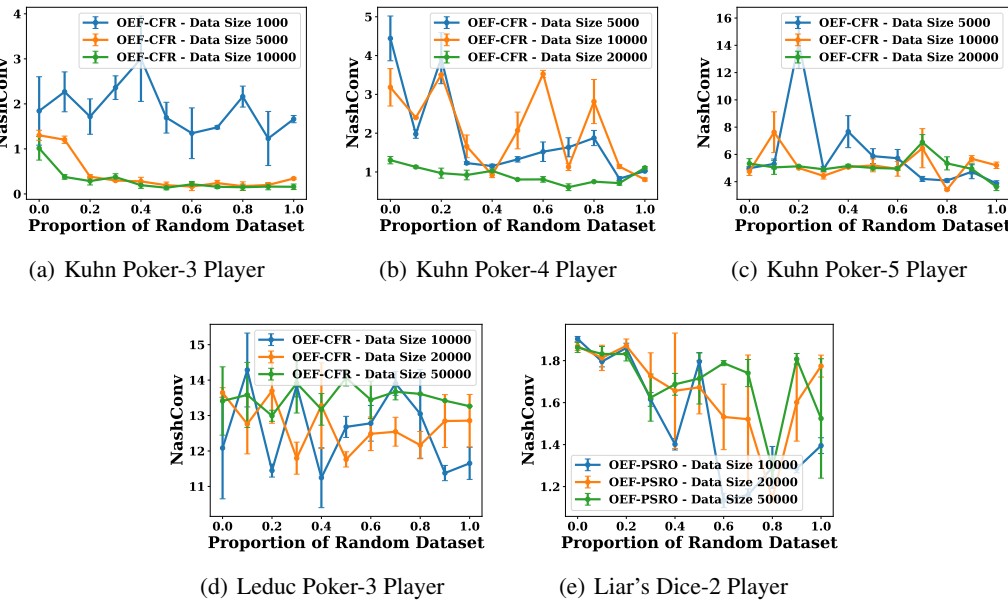

(a) Kuhn Poker-3 Player

(b) Kuhn Poker-4 Player

(c) Kuhn Poker-5 Player

(d) Leduc Poker-3 Player

(e) Liar's Dice-2 Player

Figure 12: Experimental results for the MB method

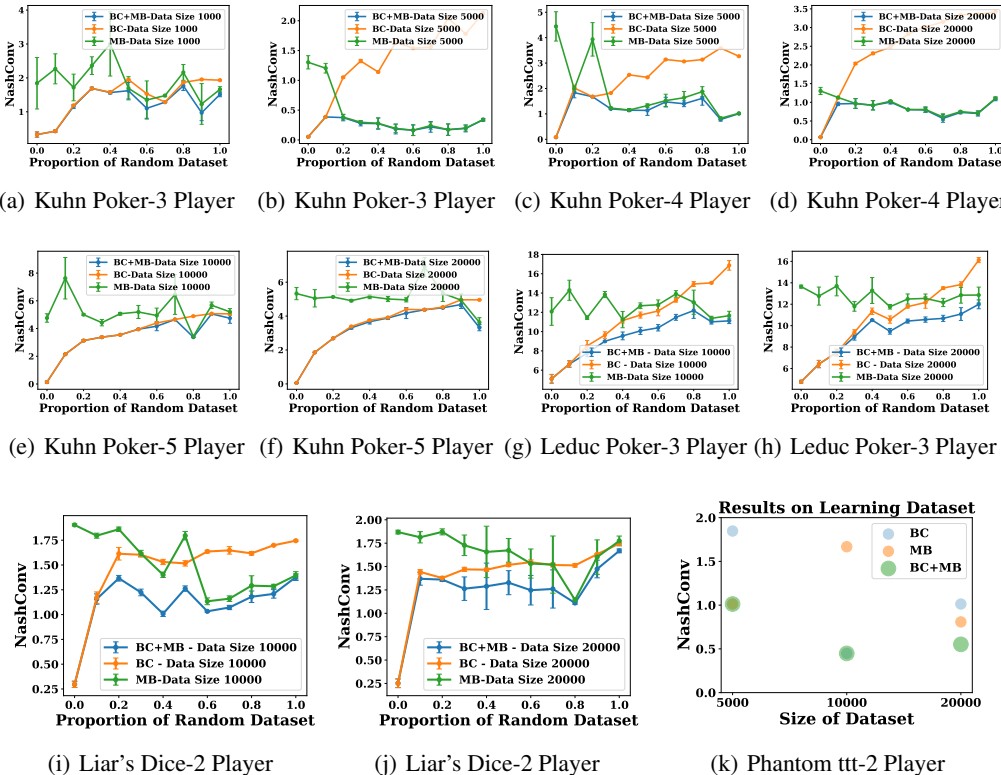

(a) Kuhn Poker-3 Player  (b) Kuhn Poker-3 Player  (c) Kuhn Poker-4 Player  (d) Kuhn Poker-4 Player

(e) Kuhn Poker-5 Player  (f) Kuhn Poker-5 Player  (g) Leduc Poker-3 Player  (h) Leduc Poker-3 Player

(i) Liar's Dice-2 Player  (j) Liar's Dice-2 Player  (k) Phantom ttt-2 Player

Figure 13: Experimental results for the benchmark algorithm BC+MB

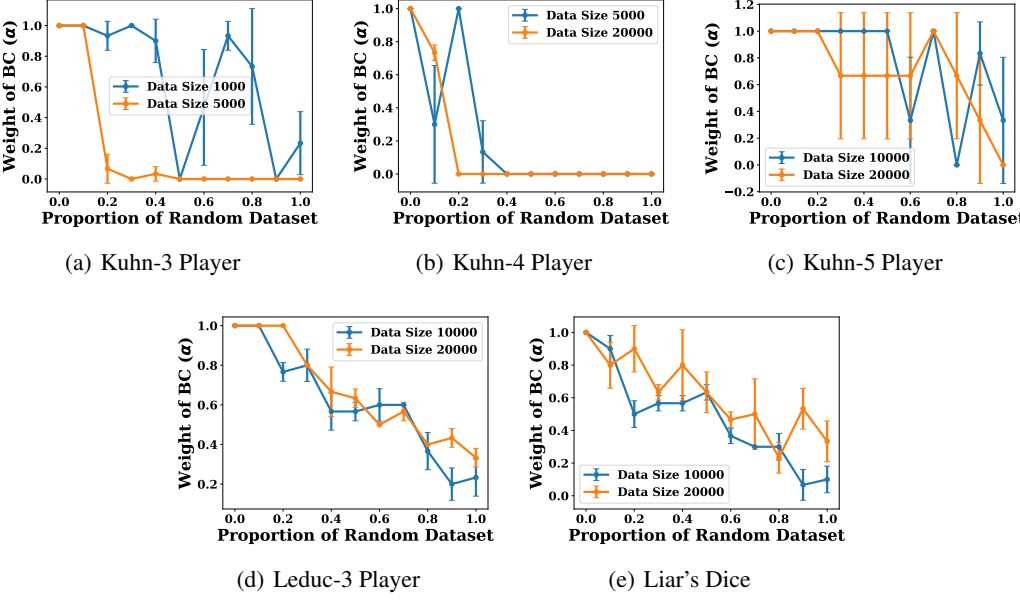

(a) Kuhn-3 Player  (b) Kuhn-4 Player  (c) Kuhn-5 Player

(d) Leduc-3 Player  (e) Liar's Dice

Figure 14: Experimental Results for Proper Weight

**Ablation Study.** To figure out the influence of hyperparameters, we conduct some ablation experiments on two-player Kuhn poker and Leduc poker games. We consider different model structures with various hidden layers. Specifically, for the 2-Player Kuhn poker game, we use different environment models with 8, 16, 32, and 64 hidden layers. For the 2-Player Leduc poker game, which is a more complicated game, the numbers of hidden layers for different models are 32, 64, and 128. Besides, we train the environment models for different epochs to evaluate the robustness of our approach. Figures 15-16 show these ablation results. We can find that the hidden layer size and the number of train epochs have little effect on the performance of the BC algorithm. These results further verify that the performance of the BC algorithm mainly depends on the quality of the dataset. As we know that the performance of the model-based algorithm mainly depends on the trained environment model. Since the size of the hidden layer and the number of train epochs will influence the training phase of the environment model, the size of the hidden layer and the number of train epochs have a slight influence on the performance of the model-based algorithm. As long as the size of the hidden layer and the number of train epochs can guarantee that the model is well-trained, the performance of the model-based algorithm will not be affected.

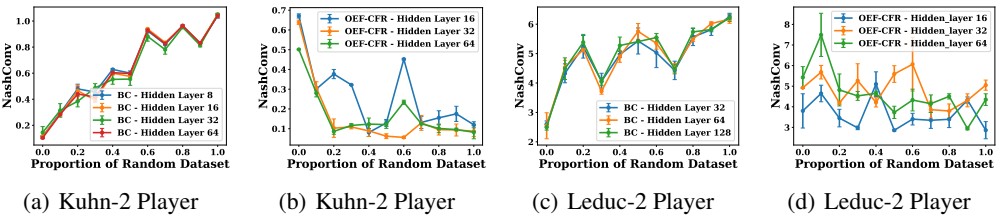

(a) Kuhn-2 Player  (b) Kuhn-2 Player  (c) Leduc-2 Player  (d) Leduc-2 Player

Figure 15: Ablation results for different hidden layer size

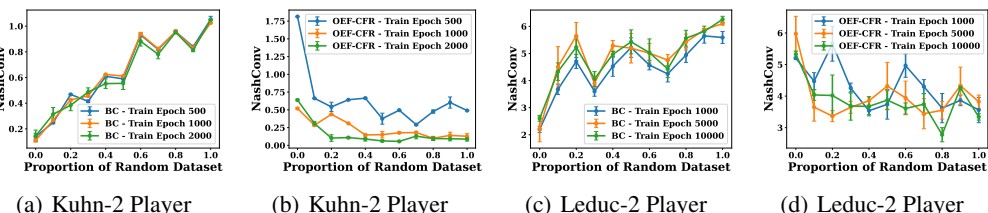

(a) Kuhn-2 Player  (b) Kuhn-2 Player  (c) Leduc-2 Player  (d) Leduc-2 Player

Figure 16: Ablation results for different train epoch

**Parameter Setting.** We list the parameters used to train the behavior cloning policy and environment model for all games used in our experiments in Table 1 and Table 2.

Table 1: Parameters for Behavior Cloning algorithm

| Games | Data size | Hidden layer | Batch size | Train epoch |
|---|---|---|---|---|
| 2-player Kuhn poker | 500 | 32 | 32 | 1000 |
| 2-player Kuhn poker | 1000 | 32 | 32 | 2000 |
| 2-player Kuhn poker | 5000 | 32 | 32 | 2000 |
| 3-player Kuhn poker | 1000 | 32 | 32 | 5000 |
| 3-player Kuhn poker | 5000 | 32 | 32 | 5000 |
| 3-player Kuhn poker | 10000 | 64 | 128 | 5000 |
| 4-player Kuhn poker | 5000 | 64 | 64 | 5000 |
| 4-player Kuhn poker | 10000 | 64 | 128 | 5000 |
| 4-player Kuhn poker | 20000 | 64 | 128 | 5000 |
| 5-player Kuhn poker | 5000 | 64 | 64 | 5000 |
| 5-player Kuhn poker | 10000 | 64 | 128 | 5000 |
| 5-player Kuhn poker | 20000 | 64 | 128 | 5000 |
| 2-player Leduc poker | 10000 | 128 | 128 | 10000 |
| 2-player Leduc poker | 20000 | 128 | 128 | 10000 |
| 2-player Leduc poker | 50000 | 128 | 128 | 10000 |
| 3-player Leduc poker | 10000 | 128 | 128 | 10000 |
| 3-player Leduc poker | 20000 | 128 | 128 | 10000 |
| 3-player Leduc poker | 50000 | 128 | 128 | 10000 |
| Liar's Dice | 10000 | 64 | 64 | 5000 |
| Liar's Dice | 20000 | 64 | 128 | 5000 |
| Liar's Dice | 50000 | 64 | 128 | 5000 |
| Phantom Tic-Tac-Toe | 5000 | 128 | 128 | 5000 |
| Phantom Tic-Tac-Toe | 10000 | 128 | 128 | 5000 |
| Phantom Tic-Tac-Toe | 20000 | 128 | 128 | 5000 |

Table 2: Parameters for training Environment Model

| Games | Data size | Hidden layer | Batch size | Train epoch |
|---|---|---|---|---|
| 2-player Kuhn poker | 500 | 32 | 32 | 1000 |
| 2-player Kuhn poker | 1000 | 32 | 32 | 2000 |
| 2-player Kuhn poker | 5000 | 32 | 32 | 2000 |
| 3-player Kuhn poker | 1000 | 32 | 32 | 2000 |
| 3-player Kuhn poker | 5000 | 32 | 32 | 5000 |
| 3-player Kuhn poker | 10000 | 64 | 128 | 5000 |
| 4-player Kuhn poker | 5000 | 64 | 64 | 5000 |
| 4-player Kuhn poker | 10000 | 64 | 128 | 5000 |
| 4-player Kuhn poker | 20000 | 64 | 128 | 5000 |
| 5-player Kuhn poker | 5000 | 64 | 64 | 5000 |
| 5-player Kuhn poker | 10000 | 64 | 128 | 5000 |
| 5-player Kuhn poker | 20000 | 64 | 128 | 5000 |
| 2-player Leduc poker | 5000 | 64 | 64 | 5000 |
| 2-player Leduc poker | 10000 | 64 | 64 | 5000 |
| 2-player Leduc poker | 20000 | 128 | 128 | 10000 |
| 3-player Leduc poker | 10000 | 128 | 128 | 10000 |
| 3-player Leduc poker | 20000 | 128 | 128 | 10000 |
| 3-player Leduc poker | 50000 | 128 | 128 | 10000 |
| Liar's Dice | 10000 | 64 | 64 | 5000 |
| Liar's Dice | 20000 | 64 | 128 | 5000 |
| Liar's Dice | 50000 | 64 | 128 | 5000 |
| Phantom Tic-Tac-Toe | 5000 | 128 | 128 | 5000 |
| Phantom Tic-Tac-Toe | 10000 | 128 | 128 | 5000 |
| Phantom Tic-Tac-Toe | 20000 | 128 | 128 | 5000 |

