# OpenReview forum: "Offline Equilibrium Finding"
_ICLR.cc/2023/Conference — Submitted to ICLR 2023_

### Official Review · Reviewer_APEF · 2022-10-20

**Confidence:** 3
**Correctness:** 2
**Technical Novelty And Significance:** 2
**Empirical Novelty And Significance:** 2
**Recommendation:** 5

**Clarity, Quality, Novelty And Reproducibility:**

This paper offers a novel approach to game-solving without any access to a simulator. I have not seen previous work in this space using modern machine learning techniques but would be surprised if this is unstudied in the field of Game Theory.

The paper contains a sizeable appendix that provides details to parts of their implementation. However, there are many moving parts across all the different ablations and baselines, and I suspect without releasing their code the results could not be reproduced.

**Strength And Weaknesses:**

**Strengths**
 - Their approach is intuitively simple, in a good way, and sensible for solving a real problem.
 - The authors include a nice set of diverse baselines to understand the limitations of most current approaches including BC and offline model-based RL, alongside combinations of them.
 - The authors perform a nice sensitivy analysis to the quality of the offline data across all methods.
 - The problem is introduced and framed in English well such that a reader from any of the diverse positions of background could understand well the main point they're investigating. Despite this, I would recommend condensing this as it resulted in less technical details from being included in the main body of the paper.

**Weaknesses**
 - I do not believe they've correctly positioned this paper within the literature. This work is an example of an EGTA algorithm sat at the extrema of the algorithmic design space: a high-fidelity game-model is constructed alongside allowing no further queries to the simulator (that is exact). The authors claim that EGTA is precluded from being used in this case, but at the core of their work is a game-model that's being solved through game-theory, which is precisely EGTA. I am further skeptical of the claim "offline RL is not enough since it can only learn the best strategy for one agent independently", could the authors explain or cite evidence as to why this must be true? Moreover, I'm surprised at the lack of discussion on behavioural cloning. If the demonstration data is exactly displaying an equilibrium then there is no issue; whereas, the more off-equilibrium the demonstration data, the more this approach matters alongside the difficulty of the correction.
 - The abstract claims a formal introduction to the proposed problem of offline equilibrium finding; however, the text contains only an informal one-sentence definition. The paper could be improved by formally defining the problem and characterizing interesting properties or questions within the space (e.g., approximation errors, selecting a single equilibrium when many exist, etc.).
 - The success of their method (game-model, then solving) strongly depends on the state-action space covered during demonstration versus the space covered by the desired solution. This is because the game-model cannot reasonably exhibit entirely novel transitions. I raise this point because in the games demonstrated in this paper it is possible that the state-action space is exceptionally well covered. I would be curious to know how this approach works outside of this domain and wonder if the authors have investigated this at all.


**Summary Of The Paper:**

This paper looks at the problem of applying offline RL to learning solutions to games. The complication that arises is that the demonstration data may not exhibit a solution to the game, and as we're offline we cannot collect new equilibrium demonstrations. This question that is investigated in this paper is exactly this: how can one learn an equilibrium in an offline fashion. They propose an algorithmic solution that is a simple two-step approach: learn a game-model, and apply a game-solving algorithm on the game-model as an approximation solution to the underlying game. They empirically demonstration their approach on several small games.

**Summary Of The Review:**

This work demonstrated that learned game-models can be solved as a proxy for the true game in an offline setting (wrt. the true game). The paper is framed as formally introducing a new problem (ie., offline equilibrium finding), but I found this component of the exposition absent. Moreover, the empirical results are all in an exceptionally idealized setting for the method: the equilibrium state-action is likely covered by any arbitrarily bad demonstration profile. Despite these concerns, the work contains a straightforward/trivial (in a good way) approach to a reasonable problem and provides a nice set of preliminary experiments in the space.

In its current state due to unsupported/incorrect claims and misaligned problem framing I would not advocate for its publication, but with moderate revisions to the text believe it could be ready.

---

> ### Author Response · Authors · 2022-11-18
> **Response to Reviewer APEF (1/2)**
>
> Thank you for the constructive comments and advice. Our detailed responses are shown below:
>
> **Question 1: This work is an example of an EGTA algorithm sat at the extrema of the algorithmic design space.**
>
> Thank you for your comment and sorry for the confusion.
> Actually, our work is different from EGTA which can be explained from two aspects.
>
> 1) As described in the previous work [1], EGTA takes the game simulator as the fundamental input and performs strategic reasoning through interleaved simulation and game-theoretic analysis. Therefore, **the game simulator is required in EGTA.** In contrast, under the OEF setting, only the offline dataset is available and the game simulator is not required.
>
> 2) The estimated game model (empirical game) in EGTA is built based on the simulation's results, which are obtained by performing **known strategies** on the simulator. In contrast, in the OEF setting, the offline dataset is generated with an **unknown strategy**. In our work, although we use different behavior strategies to generate several offline datasets, we did not utilize these behavior strategies when performing our OEF algorithm.
> Therefore, our proposed approach is different from EGTA. It is more challenging to find the equilibrium strategy in our OEF setting.
>
> We have added a detailed discussion about why EGTA cannot work for the OEF setting and the difference between EGTA and our method in the revision (Sec. 2).
>
> [1] Wellman M P. Methods for empirical game-theoretic analysis[C]//AAAI. 2006: 1552-1556.
>
> ---
>
> **Question 2: I am further skeptical of the claim "offline RL is not enough since it can only learn the best strategy for one agent independently", could the authors explain or cite evidence as to why this must be true?**
>
> Thank you for your interesting question.
> Offline single-agent RL algorithms are designed to learn a policy to achieve the highest expected value from static datasets of previously collected interactions. In other words, offline single-agent RL can learn the best response strategy for one individual player.
> In the OEF setting, our goal is to find the equilibrium strategy for the underlying game with multiple players. Although under the equilibrium strategy, every player plays their best response strategy, using the offline single-agent RL algorithm to learn the player's best response strategy independently may not converge to the equilibrium strategy.
> Currently, offline multi-agent RL algorithms mainly focus on solving cooperative settings which is not suitable for our competitive setting.
> In the experiment part, we conduct several offline RL algorithms, and these results show that offline RL algorithms cannot converge to the equilibrium strategy effectively.
>
> ---
>
> **Question 3: Discussion on behavioural cloning. If the demonstration data is exactly displaying an equilibrium then there is no issue;  whereas, the more off-equilibrium the demonstration data, the more this approach matters alongside the difficulty of the correction.**
>
> Correct, when the demonstration data does not display an equilibrium, the performance of the behavior cloning technique is worse. As described in Section 5.3, our model-based method can achieve satisfactory performance under the random dataset. While for the expert dataset, the model-based cannot learn a well-performed model. Therefore, we apply the behavior cloning technique to improve the performance under the expert dataset.
> Experimental results show that our BC+MB algorithm can perform well under all these datasets.
>
> ---
>
> **Question 4: The abstract claims a formal introduction to the proposed problem of offline equilibrium finding; however, the text contains only an informal one-sentence definition. The paper is framed as formally introducing a new problem (ie., offline equilibrium finding), but I found this component of the exposition absent.**
>
> Thanks for your suggestions. We have provided a formal definition of OEF in Section 4 of the revision.

---

> > ### Author Response · Authors · 2022-11-18
> > **Response to Reviewer APEF (2/2)**
> >
> > **Question 5: The success of their method (game-model, then solving) strongly depends on the state-action space covered during demonstration versus the space covered by the desired solution. I would be curious to know how this approach works outside of this domain and wonder if the authors have investigated this at all. The empirical results are all in an exceptionally idealized setting for the method: the equilibrium state-action is likely covered by any arbitrarily bad demonstration profile.**
> >
> > Yes, the state-action space coverage affects the final performance. In this paper, we proposed four types of datasets: expert dataset, learning dataset, random dataset, and hybrid dataset. The main difference among these datasets is the state-action space coverage. We added some theoretical analysis of the influence of data coverage on our OEF algorithm (Appendix B.1) in the revision. The performance of the model-based algorithm decreases when the data coverage is limited (i.e., the expert dataset).
> > To address this issue, we propose to apply the behavior cloning technique to get a good policy under the expert dataset.
> > Therefore, our combined algorithm BC+MB can perform well under different data coverage.
> >
> > ---
> >
> > **Question 6: However, there are many moving parts across all the different ablations and baselines, and I suspect without releasing their code the results could not be reproduced.**
> >
> > Thanks for your comment. We submitted the source code as supplementary materials. We can also release our code for researchers to reproduce our results.

---

### Official Review · Reviewer_2VY7 · 2022-10-22

**Confidence:** 3
**Correctness:** 4
**Technical Novelty And Significance:** 2
**Empirical Novelty And Significance:** 2
**Recommendation:** 5

**Clarity, Quality, Novelty And Reproducibility:**

The paper is very clearly written, and seems to solve a new and interesting problem. The main ideas of the paper, though, are quite straightforward, and as such I consider the novelty of the paper to be its weakest point.

Code to reproduce experiments is provided. I have not run the code.

**Strength And Weaknesses:**

I find the premise of the paper interesting--offline RL is of obvious interest, and applying it to games causes additional issues as the authors rightfully point out. For example, the fact that independent offline RL (e.g., the tested algorithms MOPO and BAIL) fails in this setting is fairly clear: given strategy profile $\sigma$ in an extensive-form game, even if each player $i$ builds a perfect environment (i.e., game + opponent) model and independently computes a perfect strategy in that model (i.e., a best response $\sigma_i'$ to $\sigma_{-i}$), it would be quite surprising if somehow the profile $\sigma'$ turned out to be close to equilibrium. Thus, other methods are required.

My main issue is that the paper is very light on theory and new ideas. I concede that to some extent this is my own theoretical background seeping through, but in any case I found the technical section of the paper to be straightforward. It seems to boil down to: "to solve a game you don't know, first approximate the game and then solve the approximated game", applying known techniques in turn to each of the two subproblems.

The experimental section of the paper is fairly robust, but basically verifies what one would expect from the methods tested, and to me contained no surprises. I would also like to see larger experiments--perhaps in a game large enough that directly computing an equilibrium is hard, and in which the BC+MB method outperforms pure BC.

Questions:

1. Why should we expect a Fourier transform of node frequencies to be illuminating? Are the nodes ordered in some meaningful way such that, say, the "amplitude at frequency 2" is a meaningful thing?

1. I did not understand Figures 4(h) and 5(h). In particular, what's a "player" for those figures? Aren't Kuhn and Leduc both two-player games?

1. In the experiments in Section 5.4, how was a Nash equilibrium computed for the purpose of collecting the OEF dataset? The games seem too large to use an actual algorithm that is guaranteed to compute general-sum Nash.

1. The experiments in the paper are all with approximate equilibrium finders (namely, based on deep learning), despite the fact that some of the games are small enough that better equilibrium finders (e.g., tabular CFR+) could have been used instead. Given this, to what extent are the experimental observations dependent on the quality of the equilibrium found vs. the quality of the approximated game model? For example, the NashConv values in (2p) Leduc poker seem rather large, and my first guess would be that this is due to equilibrium finder failure--Leduc poker is a fairly simple game, so I would guess that 20000 data points is plenty to build a decent game model.

1. How is $\alpha$ selected for BC+MB? In particular, you state that $\alpha$ is selected by "testing the final policies in a real game to get the best final policy"--against what opponent(s) were the policies optimized?



**Summary Of The Paper:**

The authors propose offline equilibrium finding, which can essentially be summarized as a method of equilibrium finding in unknown games using only a training set of gameplay--by first building an approximate model of the game from the training set and then computing an equilibrium of the model. Experiments are run to demonstrate the efficacy of the method.

**Summary Of The Review:**

An interesting concept, but I'm not sure of novelty, as the main ideas of the paper are fairly natural and not analyzed very deeply.

---

> ### Author Response · Authors · 2022-11-18
> **Response to Reviewer 2VY7 (1/2)**
>
> Thank you for the constructive comments and advice. Our detailed responses are shown below:
>
> **Question 1: My main issue is that the paper is very light on theory and new ideas. I concede that to some extent this is my own theoretical background seeping through, but in any case I found the technical section of the paper to be straightforward.**
>
> Thanks for your comments!
> Our work focuses on solving a new problem, Offline Equilibrium Finding, whose goal is to find an equilibrium strategy based on an offline dataset only. It is challenging to build the relationship between the equilibrium strategy and the offline dataset. Therefore, we propose a general model-based framework to fill this gap. We want to emphasize the challenges and the contribution of our work as follows:
>
> **Firstly, we propose a new research problem: Offline Equilibrium Finding (OEF).** Given an offline dataset collected using an unknown behavior strategy in a game, our goal is to find an approximate equilibrium strategy profile of the game.
> It is challenging to solve such a problem since we need to build a relationship between the offline dataset and the equilibrium strategy.
>
> **Secondly, we construct four types of OEF datasets: random dataset, expert dataset, learning dataset and hybrid dataset.** In this paper, we collect datasets from several widely used games including two-player games (*Kuhn* and *Leduc*, *Phantom Tic-Tac-Toe*, and *Liar’s Dice*) and multi-player games (*Kuhn* and *Leduc*).
>
> To solve the OEF problem, existing algorithms, such as offline RL, and opponent modeling, cannot work as introduced in section 2 of the paper, and directly learning the equilibrium strategy from the offline dataset is very challenging. Therefore, **we propose a novel model-based OEF framework by introducing an environment model.** The environment model is trained based on the offline dataset and then used for computing the equilibrium strategy. Therefore, it can be the intermediary between the offline dataset and the equilibrium strategy. The proposed framework is general and can be applied to any online equilibrium finding algorithms to solve the OEF problem.
>
> Since the performance of the model-based algorithm under the expert dataset is unsatisfactory, **to further improve the performance, we propose to combine the behavior cloning technique.** Behavior cloning can mimic the behavior strategy from an offline dataset. The cloned strategy can be mixed with the strategy learned using the model-based algorithm by assigning different weights to them to get a better strategy.
>
> Furthermore, we provided some theoretical analysis of the influence of dataset coverage on the algorithm (Appendix B.1) in the revision.
>
> ---
>
> **Question 2: I would also like to see larger experiments--perhaps in a game large enough that directly computing an equilibrium is hard, and in which the BC+MB method outperforms pure BC.**
>
> Our model-based offline method is adapted from online equilibrium finding algorithms to solve the OEF problem (e.g., CFR and PSRO).
> In the scenario that we do not have an accurate simulator, given an offline dataset, our model-based method can compute a satisfying equilibrium strategy efficiently.
> To some extent, our method depends on the performance of online equilibrium-finding algorithms.
> If the game is hard to directly compute the equilibrium strategy, it means that the existing online equilibrium-finding algorithm cannot solve it effectively.
> For large-scale games that cannot be solved by online algorithms, the offline algorithm may be unavailable either since the computing resources required may be very large.
> But we want to highlight that our model-based framework is more efficient for the offline equilibrium finding problems.
>
> ---
>
> **Question 3: Why should we expect a Fourier transform of node frequencies to be illuminating? Are the nodes ordered in some meaningful way such that, say, the "amplitude at frequency 2" is a meaningful thing?**
>
> Yes, the nodes are arranged in a meaningful way, i.e., depth-first search results, as we introduced in Appendix C. We want to show that the distribution of different datasets is quite different. And the results also reflect the dataset coverage. These differences may result in different performance for BC and MB methods.

---

> > ### Author Response · Authors · 2022-11-18
> > **Response to Reviewer 2VY7 (2/2)**
> >
> > **Question 4: What's a "player" for those figures? Aren't Kuhn and Leduc both two-player games?**
> >
> > Sorry for the confusion. "player" for those figures represents the number of players in the game.
> > There are different types of Kuhn and Leduc games, including two-player ones and multi-player ones [1,2]. In our experiments, we use both the two-player and multi-player games. We have improved the description to make it clear in the revision.
> >
> > [1] Kuhn poker, https://en.wikipedia.org/wiki/Kuhn_poker
> >
> > [2] Marris L, Muller P, Lanctot M, et al. Multi-agent training beyond zero-sum with correlated equilibrium meta-solvers[C]//International Conference on Machine Learning. PMLR, 2021: 7480-7491.
> >
> > ---
> >
> > **Question 5: In the experiments in Section 5.4, how was a Nash equilibrium computed for the purpose of collecting the OEF dataset? The games seem too large to use an actual algorithm that is guaranteed to compute general-sum Nash.**
> >
> > In the experiments in Section 5.4, the dataset used is the same as that used in computing the NE strategy, i.e., OEF datasets. CFR or PSRO is used to compute the Nash equilibrium strategy to collect the dataset. Due to the large game size, both CFR and PSRO cannot solve the game efficiently. Therefore, the Nash equilibrium strategy is an approximation solution, not an exact equilibrium strategy.
> >
> > ---
> >
> > **Question 6: The experiments in the paper are all with approximate equilibrium finders (namely, based on deep learning), despite the fact that some of the games are small enough that better equilibrium finders (e.g., tabular CFR+) could have been used instead. Given this, to what extent are the experimental observations dependent on the quality of the equilibrium found vs. the quality of the approximated game model?**
> >
> > Thanks for your interesting question!
> > Our model-based offline equilibrium finding framework consists of two key components, the environment model and the online equilibrium finding algorithm.
> > Therefore, the performance of our approach depends on the quality of both the game model and the equilibrium found.
> >
> > 1) Take the Kuhn poker game as an example since it is small that any equilibrium finder can get the approximate or accurate equilibrium strategy. As shown in Figure 4(d), we can find OEF-CFR and OEF-PSRO have similar performance based on the same approximated game model, i.e., under the same dataset. However, the performance of these algorithms decreases as the proportion of random datasets decreases. It means that as the quality of the approximated game model decreases, the performance of OEF-CFR/OEF-PSRO becomes worse. It indicates that if the equilibrium finder can effectively approximate the equilibrium strategy, then the performance of the model-based algorithm mainly depends on the quality of the approximated game model.
> >
> > 2) On the other hand, we consider the quality of the approximated game model trained based on the random dataset as a perfect game model of the actual game. From the results in Figure 5(d) and Figure (12), we can find that in some large games, the performance of OEF-CFR/OEF-PSRO is not satisfactory. It may be caused by the online equilibrium finding algorithms, which means that the performance of the online equilibrium finding algorithm can also influence the performance of our model-based algorithm. Nevertheless, it is not the focus of our work. Our main contribution is proposing a general model-based framework that bridges the relationship between the offline dataset and the equilibrium finding algorithms.
> >
> > ---
> >
> > **Question 7: How is $\alpha$ selected for BC+MB?**
> >
> > Sorry for the confusion. The choice of $\alpha$ depends on the exploitability, i.e., the closeness to the Nash equilibrium strategy. We have improved the description of this setting in the revision (Sec. 5.3).

---

> > > ### Comment · Reviewer_2VY7 · 2022-11-18
> > > **Response**
> > >
> > > Thank you for your response. The response has not significantly changed my view of the paper, and thus I do not alter my score.
> > >
> > > Here are some concrete suggestions for a future revision.
> > >
> > > * For the small games used in experiments, I would advise moving to a high-precision, tabular equilibrium finding algorithm such as tabular CFR+ or linear programming. It will allow the isolation of the learned game model as an experimental variable.
> > > * I do not agree with Theorem 2 (Appendix B.1). If the offline dataset is generated from an equilibrium profile $\pi$, the dataset would only be computed from *on-path* equilibrium play. Thus, behavioral cloning may not successfully learn how an agent should *counterfactually* play after the opponent plays an action never played under $\pi$---that is, how to correctly punish mistakes---even with an infinite dataset. Thus, the resulting strategy may be exploitable.
> > > * More broadly, it may be interesting to shore up the theoretical grounding of the paper, especially about the main proposed algorithm BC+MB. For example, in an arbitrary IIG, given a dataset generated from a profile $\pi$, what conditions are needed so that it is possible to recover a profile with exploitability at least as good as $\pi$? How about better? By the previous bullet, even assuming $\pi$ is an equilibrium profile may not be enough.
> > > * Re. Q5 (Nash equilibrium computation): CFR+ does not converge to Nash equilibria in general-sum games, even in the tabular setting--it converges to normal-form coarse-correlated equilibrium (NFCCE). Using it as an algorithm to compute Nash equilibria in general-sum games is thus incorrect. As for PSRO, the question still remains: how is the equilibrium of the meta-game computed? If that is by CFR+, again that algorithm will only reach NFCCE.

---

> > > > ### Author Response · Authors · 2022-11-19
> > > > **Response**
> > > >
> > > > Thanks for your reply. For your concern, we provide some explanation shown below.
> > > >
> > > > **Q1: For the small games used in experiments, I would advise moving to a high-precision, tabular equilibrium finding algorithms such as tabular CFR+ or linear programming. It will allow the isolation of the learned game model as an experimental variable.**
> > > >
> > > > Under the OEF setting, its goal is to find an equilibrium strategy only based on a fixed offline dataset collected by an unknown behavior strategy. Our model-based algorithm first trains an environment model (neural network) based on the offline dataset. Then we can compute the equilibrium strategy of the game based on the environment model.
> > > > However, it would cause some issues when applying tabular-based online equilibrium-finding algorithms. In some cases, the trained environment model may produce different state transition information from the actual game since some state transitions may not be in the offline dataset. Therefore, under this case, if we apply the tabular-based algorithm, the policy represented by a table would not be applied to the actual game since some states of the actual game are not in the table. Therefore, we choose deep-learning-based algorithms, which have a more powerful generalization ability to unseen states, to compute the equilibrium strategy.
> > > >
> > > > ---
> > > >
> > > > **Q2: I do not agree with Theorem 2 (Appendix B.1). If the offline dataset is generated from an equilibrium profile $\pi$, the dataset would only be computed from on-path equilibrium play. Thus, behavioral cloning may not successfully learn how an agent should counterfactually play after the opponent plays an action never played under $\pi$ ---that is, how to correctly punish mistakes---even with an infinite dataset. Thus, the resulting strategy may be exploitable.**
> > > >
> > > > If the offline dataset is generated from an equilibrium profile $\pi$ and the behavior cloning policy is well-trained (precisely mimics the behavior strategy used to generate the offline dataset), then the behavior cloning policy $\pi^{bc}$ is the same as equilibrium profile $\pi$.
> > > > Since $\pi$ is the equilibrium strategy, then $\pi$ is the strategy that achieves zero exploitability. It means that whatever strategy the opponent plays, the equilibrium strategy would not perform worse than the case that the opponent plays the equilibrium strategy. Therefore, $\pi^{bc}$ has the same performance as the equilibrium strategy $\pi$, i.e., it will not be exploited.
> > > >
> > > > ---
> > > >
> > > > **Q3: More broadly, it may be interesting to shore up the theoretical grounding of the paper, especially about the main proposed algorithm BC+MB.**
> > > >
> > > > Thanks for your valuable suggestion. We added more theoretical analysis about BC+MB in the version (Appendix B.1). More specifically, we provide several additional theorems (i.e., Theorem 3-6) about under what condition on the behavior strategy $\sigma$, our algorithm BC+MB can get the strategy that performs equal to or better than the behavior strategy $\sigma$ used to generate the offline dataset.
> > > >
> > > > ---
> > > >
> > > > **Q4: Re. Q5 (Nash equilibrium computation): CFR+ does not converge to Nash equilibria in general-sum games, even in the tabular setting--it converges to normal-form coarse-correlated equilibrium (NFCCE). Using it as an algorithm to compute Nash equilibria in general-sum games is thus incorrect. As for PSRO, the question still remains: how is the equilibrium of the meta-game computed? If that is by CFR+, again that algorithm will only reach NFCCE.**
> > > >
> > > > Thanks for your interesting question, and sorry for the confusion. Indeed, both CFR and PSRO cannot converge to NE in general-sum games. But, in our approach, CFR or PSRO is only used to collect the data in those multi-player poker games. Even though they cannot guarantee to compute a Nash equilibrium in general sum games, they are useful in getting good strategies to collect data. For example, PSRO with $\alpha$-rank as the meta solver can get a strategy with low exploitability under general-sum many-player games and CFR-based algorithm [1] can also get a superhuman strategy in the multi-player poker game. We have clarified this point in the revision (Appendix B).
> > > >
> > > > [1] Brown N, Sandholm T. Superhuman AI for multiplayer poker[J]. Science, 2019, 365(6456): 885-890.

---

### Official Review · Reviewer_XR74 · 2022-10-23

**Confidence:** 4
**Correctness:** 4
**Technical Novelty And Significance:** 3
**Empirical Novelty And Significance:** 3
**Recommendation:** 6

**Clarity, Quality, Novelty And Reproducibility:**

The presentation is clear and of high quality.
The problem and model are novel.
The experimental setup is too complex to reproduce the results. The results are only reproducible with Opensourced codes.

**Strength And Weaknesses:**

Strength
1. The problem is relatively new and the method is novel.
2. The authors gave a thorough literature review of the equilibrium-finding algorithms and give a sensible discussion on why introducing the offline equilibrium finding.
3. The experimental results are also extensive.

Weakness
1. The proposed method is model-based but the environment model is hard to learn. Specifically, the dynamics of the game environment may be time-variant and stochastic and cannot easily be characterized by a DNN proposed by the authors. It is worth exploring how the OEM performs when the model is perfectly known in simulation and how its performance will be degraded when the model is not well learned.
2. In Fig 4(h), why does the NashConv not exist for MB when the number of players is 2 and 3 and the same issue for BC when the number of plays is 4 and 5? Similar question for MB when the number of players is 3 in Fig 5(h).
3. The parameters used in the OEM are only briefly discussed in the Appendix (Tables 1 for BC and 2 for the model). It will be helpful to conduct some ablation study on these parameters and demonstrate how the performance is affected by these parameters.


**Summary Of The Paper:**

The authors introduced a new problem called offline equilibrium finding and designed a model-based method to apply any online equilibrium finding algorithm to the OEF setting.

**Summary Of The Review:**

Considering the high quality and novelty of the paper, I would recommend the acceptance of this work.

----------
I have read the authors' responses and would like to keep the same score.

---

> ### Author Response · Authors · 2022-11-18
> **Response to Reviewer XR74**
>
> Thank you for the constructive comments and advice. Our detailed responses are shown below:
>
> **Question 1: How the OEF performs when the model is perfectly known in simulation and how its performance will be degraded when the model is not well learned?**
>
> Since the proposed framework is a model-based approach, the performance of OEF-CFR/OEF-PSRO mainly depends on how well the environment model is trained.
> There are several factors that affect the learning of the environment model: offline dataset, model structure, and training epochs. An ablation study about the impact of model structure and training epochs can be found in Appendix E.
> In general, as introduced in Sec. 5, the training performance of the environment model highly depends on the offline dataset. For a random dataset, the environment model is well-trained and can be regarded as perfectly known. As shown in Figures 4(d) and 5(d), OEF-CFR can get a strategy close to the equilibrium strategy under the random dataset (when the proportion of the random dataset equals 1). As the proportion of the random dataset decreases, the environment model trained derives from the actual game. It can be considered as the model is not well-learned. Thus, the gap between the trained strategy and the equilibrium strategy becomes larger and larger.
>
> ---
>
> **Question 2: In Fig 4(h), why does the NashConv not exist for MB when the number of players is 2 and 3 and the same issue for BC when the number of plays is 4 and 5? Similar question for MB when the number of players is 3 in Fig 5(h).**
>
> Thank you for pointing this and sorry for the confusion. Actually, the NashConv exists for the cases in Figures 4(h) and 5(h). However, some nodes are covered by the node of BC+MB. We have corrected these issues in the revision so that every node (results) can be seen clearly.
>
> ---
>
> **Question 3: Conduct some ablation study**
>
> Thank you so much for your constructive advice. We conducted extensive ablation studies, including different model structures and training epochs. We have added the results (Appendix E) in the revision.

---

> > ### Comment · Reviewer_XR74 · 2022-11-28
> > **Keep the same score**
> >
> > Thanks for your response.
> > For Q1, 'model is perfectly known in simulation' refers to the environment being explicitly known and might be deterministic for simplicity.
> > Before introducing randomness, it is worth proving the concept with a deterministic setting.

---

### Official Review · Reviewer_ZPQh · 2022-10-27

**Confidence:** 2
**Correctness:** 3
**Technical Novelty And Significance:** 2
**Empirical Novelty And Significance:** 2
**Recommendation:** 3

**Clarity, Quality, Novelty And Reproducibility:**

The presentation should be improved. For example, the notation of state $s$ appears in Section 4.1 for the first time, but it is not formally defined in the preliminaries. Also it should emphasized that whether $a$ is the joint-action.

**Strength And Weaknesses:**

Strength: The paper focuses on offline multi-agent reinforcement learning, which is a very challenging problem in related fields. The authors consider model-based framework for finding the equilibrium, which is even more challenging, since estimating the model in multi-agent learning problems is essentially hard.



Weaknesses:  1) The motivation is not clear. In the example in Section 2, by watching the replays of the other player $B$, the learner $A$ could only find the best response to $B$'s policy, instead of an equilibrium. I am confused that how to connect this scenario with equilibrium finding. 2) The contribution in algorithm design is incremental, where no useful insights about multi-agent learning is provided.  3) It is unfair to compare the proposed algorithm with the offline RL algorithms for single agent, since the target of those algorithms is not to find an equilibrium. 4) This paper is lack of theorectical insights. It would be much better by providing the sample complexity analysis.

**Summary Of The Paper:**

The paper studies offline multi-agent reinforcement learning. The authors propose a framework for offline equilibrium finding, where the learner first trains a model $E$ by the offline dataset, and then compute an equilibrium using the model and some proper online algorithms. The authors also consider to set the behavior cloning policy as the regularization for the online equilibrium.

**Summary Of The Review:**

Overall I think this paper is incremental, and the motivation is not clear. I consider to reject this paper.

---

> ### Author Response · Authors · 2022-11-18
> **Response to Reviewer ZPQh**
>
> Thanks for your constructive comments and advice. Our detailed responses are shown below:
>
> **Question 1: How to connect the example scenario with equilibrium finding?**
>
> In the example scenario, to obtain a larger reward, Player A tends to apply the best strategy (i.e., the best response against the previous policy of Player B). However, this best strategy may be exploited by Player B if he changes his strategy accordingly. Therefore, Player A has to learn more game information by observing the replays (e.g., actions and preferences of Player B). To avoid being exploited as much as possible, the optimal solution for Player A is to choose the Nash equilibrium strategy of the underlying game. We have added more descriptions of this example scenario in the revision.
>
> ---
>
> **Question 2: The contribution in algorithm design is incremental, where no useful insights about multi-agent learning is provided.**
>
> Although we adapted some existing online equilibrium finding algorithms, we want to emphasize the contribution of our works from four aspects.
>
> **Firstly, we propose a new research problem: Offline Equilibrium Finding (OEF).** Given an offline dataset collected using an unknown behavior strategy in a game, our goal is to find an approximate equilibrium strategy profile of the game.
> It is challenging to solve such a problem since we need to build a relationship between the offline dataset and the equilibrium strategy.
>
> **Secondly, we construct four types of OEF datasets: random dataset, expert dataset, learning dataset, and hybrid dataset.** In this paper, we collect datasets from several widely used games, including two-player games (*Kuhn*,  *Leduc*, *Phantom Tic-Tac-Toe*, and *Liar’s Dice*) and multi-player games (*Kuhn* and *Leduc*).
>
> **Thirdly, we propose a novel model-based OEF framework by introducing an environment model.** The model-based framework first trains an environment model based on the offline dataset. The environment model then is used for computing the equilibrium strategy. Therefore, it can be the intermediary between the offline dataset and the equilibrium strategy. The proposed framework is general and can be applied to any online equilibrium finding algorithm to solve the OEF problem.
>
> **Lastly, to further improve the performance, we propose to combine the behavior cloning technique.** Behavior cloning can mimic the behavior strategy from an offline dataset. The cloned strategy can be mixed with the strategy learned using the model-based algorithm by assigning different weights to them to get a better strategy. Furthermore, we provided some theoretical analysis of the influence of dataset coverage on the algorithm (Appendix B.1) in the revision.
>
> We believe that our efforts can provide some insights to researchers in large-scale equilibrium finding for competitive multi-agent settings.
>
> ---
>
> **Question 3: It is unfair to compare with the offline RL for single agent.**
>
> We agree that the single-agent offline RL algorithm is not designed to find an equilibrium strategy.
> However, under the offline equilibrium finding setting, the offline RL algorithm is a more reasonable way to compute the best response strategies for each individual player.
> Actually, the poor performance of the single-agent offline RL algorithm is the motivation of our approach, rather than a baseline.
> The results in our experiments are just used to clarify that the offline RL is not enough for offline equilibrium finding. Therefore, we propose a new model-based framework to solve large-scale OEF problems.
>
> ---
>
> **Question 4: Lack of theoretical insights.**
>
> As introduced in Sec.1 in our paper, a concurrent paper provided some theoretical results about the dataset coverage problem [1]. Following their work, we also provided some theoretical analysis about the influence of dataset coverage on our algorithm (Appendix B.1) in the revision.
> Our experimental results are consistent with these theoretical conclusions. And we argue that effective and practical algorithms are more urgently needed than the perfect theoretical results for the OEF problems.
>
> [1] Cui Q, Du S S. When is Offline Two-Player Zero-Sum Markov Game Solvable?[J]. arXiv preprint arXiv:2201.03522, 2022
>
> ---
>
> **Question 5: The notation of state $s$ appears in Section 4.1 for the first time, but it is not formally defined in the preliminaries. Also it should emphasized that whether $a$ is the joint-action.**
>
> Thanks for the suggestion, and sorry for the confusion. We have improved our description and made these notations clear in the revision.

---

> > ### Comment · Reviewer_ZPQh · 2022-12-04
> > **Keep the same score**
> >
> > Thank you for the response and revision. I admit that the problem is new and worh studying, bu the current version is not good enough to be published. Firstly, the algorithm is limited due to model estimation and behavior cloning, and might fail in some complicated game environments (e.g., Startcraft2); secondly, current theorectical analysis is not concrete. For example, in Definition 1, OFE is defined to be finding an equilibrium given some offline dataset, but no accuracy guarantee is provided. Also in Appendix B.1, the terminology "covered" is never formally defined, and the following proofs are hard to understand.

---

### Author Response · Authors · 2022-11-19
**To All Reviewers**

We thank all the reviewers for their valuable and insightful comments. We have uploaded a new version and highlighted these changes in red.
These changes can be summarized as follows.

1) We provided a formal definition of the OEF problem in Section 4.

2) We provided six theorems in Appendix B.1 to analyze the relationship between the data coverage and our algorithm.

3) We highlighted our challenge that it is very challenging to directly build the relationship between the dataset and the equilibrium strategy.

4) We clarified the difference between our work and empirical game theoretical analysis (EGTA).

5) We added some ablation studies on the hyperparameters in Appendix E.

6) We fixed some confusing expressions to make our paper clear.

---

### Decision · Program_Chairs · 2023-01-20

**Decision:**

Reject

**Justification For Why Not Higher Score:**

NA

**Justification For Why Not Lower Score:**

NA

**Metareview: Summary, Strengths And Weaknesses:**

This paper studies the important and challenging problem of offline learning in Markov Games. It proposed some empirical algorithms that are direct adaptations of their online learning counterpart. The main complaints from the reviewers are (1) the proposed algorithms lack algorithmic novelty (2) there are no theoretical discussions about the solvability of such a problem. Overall, I also believe that this paper is not ready for publication and encourage the authors to follow the constructive feedbacks made by the reviewers to improve their paper.